# Platelet clearance via shear-induced unfolding of a membrane mechanoreceptor

Wei Deng[1], Yan Xu[2], Wenchun Chen[1], David S. Paul[3], Anum K. Syed[1], Matthew A. Dragovich[2], Xin Liang[1], Philip Zakas[1], Michael C. Berndt[4], Jorge Di Paola[5], Jerry Ware[6], Francois Lanza[7], Christopher B. Doering[1], Wolfgang Bergmeier[3,8], X. Frank Zhang[2] & Renhao Li[1]

Mechanisms by which blood cells sense shear stress are poorly characterized. In platelets, glycoprotein (GP)Ib–IX receptor complex has been long suggested to be a shear sensor and receptor. Recently, a relatively unstable and mechanosensitive domain in the GPIbα subunit of GPIb–IX was identified. Here we show that binding of its ligand, von Willebrand factor, under physiological shear stress induces unfolding of this mechanosensory domain (MSD) on the platelet surface. The unfolded MSD, particularly the juxtamembrane 'Trigger' sequence therein, leads to intracellular signalling and rapid platelet clearance. These results illustrate the initial molecular event underlying platelet shear sensing and provide a mechanism linking GPIb–IX to platelet clearance. Our results have implications on the mechanism of platelet activation, and on the pathophysiology of von Willebrand disease and related thrombocytopenic disorders. The mechanosensation via receptor unfolding may be applicable for many other cell adhesion receptors.

[1] Aflac Cancer and Blood Disorders Center, Department of Pediatrics, Emory University School of Medicine, Atlanta, Georgia 30322, USA. [2] Department of Mechanical Engineering and Mechanics, Bioengineering Program, Lehigh University, Bethlehem, Pennsylvania 18015, USA. [3] McAllister Heart Institute, University of North Carolina, Chapel Hill, North Carolina 27516, USA. [4] Faculty of Health Sciences, Curtin University, Perth, Western Australia 6845, Australia. [5] Department of Pediatrics and Human Genetics, Genomics Program, University of Colorado School of Medicine, Aurora, Colorado 80045, USA. [6] Department of Physiology and Biophysics, University of Arkansas for Medical Sciences, Little Rock, Arkansas 72205, USA. [7] UMR_S949 INSERM, Université de Strasbourg, EFS-Alsace, Strasbourg 67065, France. [8] Department of Biochemistry/Biophysics, University of North Carolina, Chapel Hill, North Carolina 27599, USA. Correspondence and requests for materials should be addressed to R.L. (email: renhao.li@emory.edu).

The platelet, the primary blood cell involved in haemostasis and thrombosis, senses and responds to shear force generated by blood flow. Particularly, von Willebrand factor (VWF) in the plasma and glycoprotein (GP)Ib–IX–V complex on the platelet surface have long been recognized as a major ligand–receptor pair for shear sensing and reception[1]. VWF is a multi-domain multimeric protein, containing in its A1 domain a binding site for the GPIbα subunit of GPIb–IX–V[2,3]. Under static or normal flow conditions, A1 is shielded in VWF and prevented from binding to GPIbα and the platelet. On immobilization or under elevated shear stress, VWF undergoes a multitude of morphological changes, thereby exposing A1 for GPIbα binding[4,5]. How VWF responds to elevated shear stress has been under scrutiny[6]. However, the mechanism by which platelets sense and react to flow shear through GPIb–IX–V, particularly the initial shear-induced event that induces platelet signalling, has remained elusive.

GPIb–IX–V is uniquely but abundantly expressed in platelets. GPIbα is covalently linked to GPIbβ through disulfides, and together they associate tightly with GPIX to form the GPIb–IX complex[7,8]. Weakly associated with GPIb–IX, GPV is not required for complex expression, VWF binding or signalling[9,10]. GPIbα contains an N-terminal ligand-binding domain (LBD) for A1 of VWF[3]. GPIb–IX has been implicated in the genesis, activation and clearance of platelets[11–13]. However, how this complex mediates these many functions remains unclear, partly due to the uncertainty about its mode of signalling. In GPIb–IX, its LBD is separated from the rest of complex and the cell membrane by a long and extended macroglycopeptide region (Fig. 1a). It is not clear how ligand binding to the LBD transmits a signal, through the macroglycopeptide region and other membrane-proximal parts of GPIb–IX, into the platelet. Recently, a relatively unstable and mechanosensory domain (MSD) was identified between the macroglycopeptide region and the transmembrane domain of GPIbα (ref. 14). Optical tweezer-controlled pulling of recombinant A1 on the engaged GPIb–IX induced unfolding of the MSD, employing an unfolding force ∼10–20 pN (ref. 14). This unfolding force is significantly lower than the drag force exerted on a platelet under physiological shear in the vasculature[15].

Here we report that VWF engagement with GPIbα under physiological shear stress induces MSD unfolding on the platelet and signalling into the platelet. The assessment of signalling, in conjunction with earlier reports, suggests that it leads to platelet clearance. Our findings have mechanistic implications on the interplay between shear and platelets, as well as that between platelet activation and clearance.

## Results

**Physiological shear and ligand binding induce GPIb signalling.** To test whether GPIb–IX can respond to physiological shear stress and induce signalling in the platelet, we first sought to establish in the lab an experimental system in which VWF binding to GPIbα and shear stress within the physiological range ($0–25\,\mathrm{dyn\,cm}^{-2}$) could be achieved. Since many conditions under which VWF is induced to bind GPIbα are complicated and may contain elements of shear beyond the physiological range, botrocetin, a snake venom C-type lectin that induces binding of plasma VWF to platelets in the absence of shear through its simultaneous interactions with the A1 of VWF and the LBD of GPIbα[16], was used in this study (Fig. 1a; Supplementary Fig. 1). Citrated human platelet-rich plasma (PRP, ∼200 k platelets per µl) was incubated with $1\,\mathrm{\mu g\,ml}^{-1}$ botrocetin, and treated with a variable but uniform shear stress in a cone-plate viscometer for 1–5 min (Supplementary Fig. 1). Platelets were then collected and

analysed by flow cytometry. Since large-scale platelet aggregates would hamper flow analysis, calcium was not added to citrated PRP to minimize platelet aggregation, although VWF-agglutinated platelets were detectable (Supplementary Fig. 1d,e)[17]. Diluting PRP to 20 k platelets per µl by normal plasma (1:9, v/v) produced similar results (Fig. 1; Supplementary Fig. 2). Consistent with earlier reports[18,19], only the combined treatment of botrocetin and shear stress (botrocetin/shear), but not either alone, induced significant shear-dependent increases in the intracellular calcium level and surface expression of P-selectin (Fig. 1; Supplementary Fig. 2). Without extracellular calcium, little activation of integrin αIIbβ3 was observed as expected in botrocetin/shear-treated platelets (Fig. 1)[17,19–21]. Importantly, botrocetin/shear also induced significant exposure of β-galactose as evidenced by increased *Erythrina cristagalli lectin* (ECL) binding (Fig. 1). When PRP was pretreated with Arg-Gly-Asp-Ser peptide and recalcified to 1 mM calcium, botrocetin/shear treatment induced comparable levels of platelet signalling, including increased ECL binding (Supplementary Fig. 3).

Spontaneous binding of VWF to GPIbα also occurs in many patients with type 2B von Willebrand disease (VWD)[22]. Plasma from a type 2B VWD patient who carried a mutant VWF (p.V1316M) gene was mixed 9:1 (v/v) with citrated PRP from healthy donors to a platelet count of $20\,\mathrm{k\,\mu l}^{-1}$ before undergoing uniform shear of 13 and $18\,\mathrm{dyn\,cm}^{-2}$. Compared with those without shear, VWF.V1316M-bound platelets that underwent the uniform shear treatment displayed significant exposure of β-galactose, increase in intracellular calcium and expression of P-selectin, the extents of which were comparable to those observed in botrocetin/shear-treated platelets (Fig. 1c; Supplementary Fig. 4). These results suggest that the molecular basis for the botrocetin/shear-induced effects in the platelet may be the same as that for VWF.V1316M/shear-induced ones, and thus pathologically relevant.

**Botrocetin enhances force-induced unfolding of MSD.** An optical tweezer system was utilized to assess the effect of botrocetin in modulating the force-induced unfolding of MSD in full-length GPIb–IX. In this system[14], recombinant human GPIb–IX in which the GPIX cytoplasmic domain was biotinylated was immobilized on a streptavidin bead held by a fixed micropipette, and recombinant A1 (VWF residues Asp1261–Pro1466) on another controlled by the optical laser trap. Recombinant A1 could bind the LBD and platelets spontaneously[3,14]. In each recorded contact-retraction cycle, the trapped A1-coated bead was moved into contact with the GPIb–IX-coated bead and then pulled away. On thousands of contact-retraction cycles under various pulling conditions, the lifetimes and unbinding forces of the A1/GPIb–IX bond were recorded (Fig. 2a,b). The addition of $1\,\mathrm{\mu g\,ml}^{-1}$ botrocetin to the system markedly increased both the bond lifetime and the unbinding force, consistent with previous reports that botrocetin enhances the association between A1 and LBD[16,23]. With an increased unbinding force in the presence of botrocetin, an MSD-unfolding event was observed in 68% of recorded force-extension pulling curves, which was significantly >19% occurrence rate in the absence of botrocetin (Fig. 2c). On the other hand, the MSD-unfolding force and extension was not altered by botrocetin (Fig. 2d). Overall, these results suggest that botrocetin-facilitated pulling of VWF on GPIb–IX induced MSD unfolding much more frequently, but to the same extent of unfolding.

**Shear and ligand induce MSD unfolding on the platelet.** In addition to platelet signalling, botrocetin/shear induced a shear-dependent decrease of GPIbα expression on the platelet surface (Fig. 3a,b). The addition of 5 mM EDTA or 10 µM

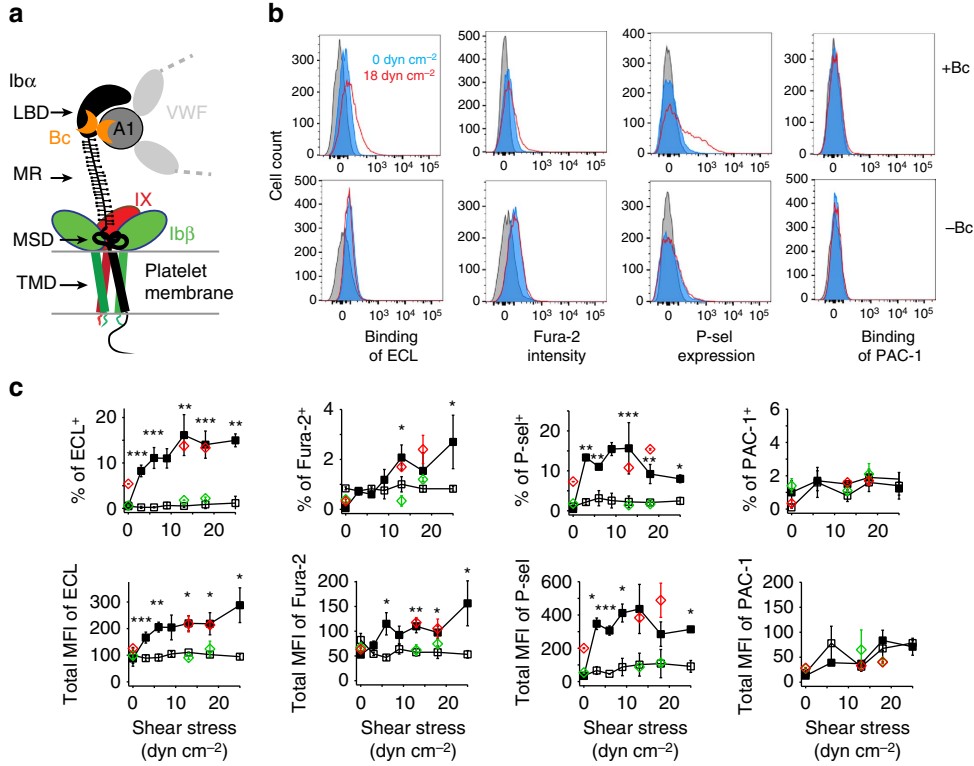

**Figure 1 | Botrocetin and physiological shear induce GPIb–IX signalling in human platelets.** (**a**) A cartoon of GPIb–IX complex illustrating botrocetin (Bc)-facilitated interaction of A1 domain of VWF with the ligand-binding domain (LBD) in GPIbα. The macroglycopeptide region (MR), the mechanosensory domain (MSD) and transmembrane domain (TMD) in GPIbα are also marked. (**b**) Representative flow histograms illustrating the effects of botrocetin and/or 18 dyn cm$^{-2}$ shear on the exposure of β-galactose (measured by binding of FITC-labelled ECL), intracellular calcium level (monitored by Fura-2 fluorescence), expression of P-selectin (binding of anti-P-selectin antibody) and activation of integrin αIIbβ3 (binding of PAC-1 antibody). Fresh human PRP (20 k platelets per μl) was mixed with or without 1 μg ml$^{-1}$ botrocetin and subjected to various uniform shear stresses. Platelets were then collected and analysed by flow cytometry for noted indicators of platelet signalling. Top row: with 1 μg ml$^{-1}$ botrocetin (+Bc); bottom row: without botrocetin (−Bc). Blue histogram: under no shear; red: under 18 dyn cm$^{-2}$ shear; grey: negative control. (**c**) Quantificative plots of platelet signalling, as either percentage of cells with noted positive signals in Supplementary Fig. 2b (top row) or median fluorescence intensity (MFI) of the entire cell population (bottom row), versus shear stress in the presence (filled squares) and absence (open squares) of botrocetin. Data are plotted as mean ± s.d. ($n = 3$). *$P < 0.05$, **$P < 0.01$, ***$P < 0.005$. Plots also include data points that were obtained from mixing type 2B VWD patient plasma (pV1316M, red diamonds) or normal plasma (green) with healthy donor platelets (1:9 v/v) under no or noted shear stress.

GM6001, a broad-spectrum metalloprotease inhibitor, before botrocetin/shear prevented the decrease, suggesting that such decrease was due to metalloprotease-mediated shedding of GPIbα (ref. 24). GPIbα is continuously shed in the platelet primarily by ADAM17 and the shedding can be upregulated when the metalloprotease becomes activated[24,25]. Botrocetin/shear did not reduce the expression levels of other ADAM17 substrates on the platelet, such as proTNF-α and GPV[26,27] (Fig. 3c,d), suggesting that botrocetin/shear induced GPIbα shedding via a mechanism that does not involve the activation of ADAM17 or other metalloproteases.

To test whether botrocetin/shear increases accessibility of the ADAM17 cleavage site in GPIbα, which is located in the MSD (Fig. 3e), fluorescein isothiocyanate (FITC)-conjugated monoclonal antibodies 5G6, WM23 and RAM.1 were mixed separately with PRP before botrocetin/shear. Whereas 5G6 binds directly the ADAM17 cleavage site (GPIbα residues 461–470), WM23 binds an epitope in the macroglycopeptide region distal to the cleavage site[28–30]. RAM.1 binds the nearby GPIbβ, which is not sheddable[31]. To simplify data interpretation, EDTA was included in the experiment to keep constant the GPIbα expression on the platelet. After botrocetin/shear, platelets were immediately fixed and antibody association measured by flow cytometry. Botrocetin/shear induced significantly more 5G6 binding to the

platelet, but little increase of WM23 or RAM.1 binding (Fig. 3f–i). Since 5G6 exhibits similar binding affinities for the isolated epitope peptide and the intact GPIb–IX[30], the observed increase in 5G6 binding reflects an increased exposure of the ADAM17 cleavage site, consistent with the unfolding of MSD under these conditions.

Postulating that on unfolding of MSD its disposition in GPIb–IX may be altered, we monitored the position of MSD relative to nearby GPIbβ on the platelet using fluorescein-conjugated 5G6 (F-5G6), fluorescein-conjugated WM23 (F-WM23), and nonfluorescent quencher-conjugated RAM.1 (Q-RAM.1) in the botrocetin/shear study. Binding of RAM.1 does not interfere with that of 5G6 (Supplementary Fig. 5a). In the absence of shear, fluorescence of the bound F-5G6, but not that of the bound F-WM23, was quenched by the bound Q-RAM.1, indicating the occurrence of specific fluorescence quenching between F-5G6 and Q-RAM.1 (Fig. 3j; Supplementary Fig. 5b). On botrocetin/shear treatment in EDTA, additional quenching of F-5G6 fluorescence by Q-RAM.1 was observed despite increased binding of F-5G6 to GPIb–IX in platelets (Fig. 3k). These results indicate a spatial change between MSD and juxtaposed GPIbβ (Supplementary Fig. 5c), providing additional evidence for botrocetin/shear-induced deformation of MSD on the platelet.

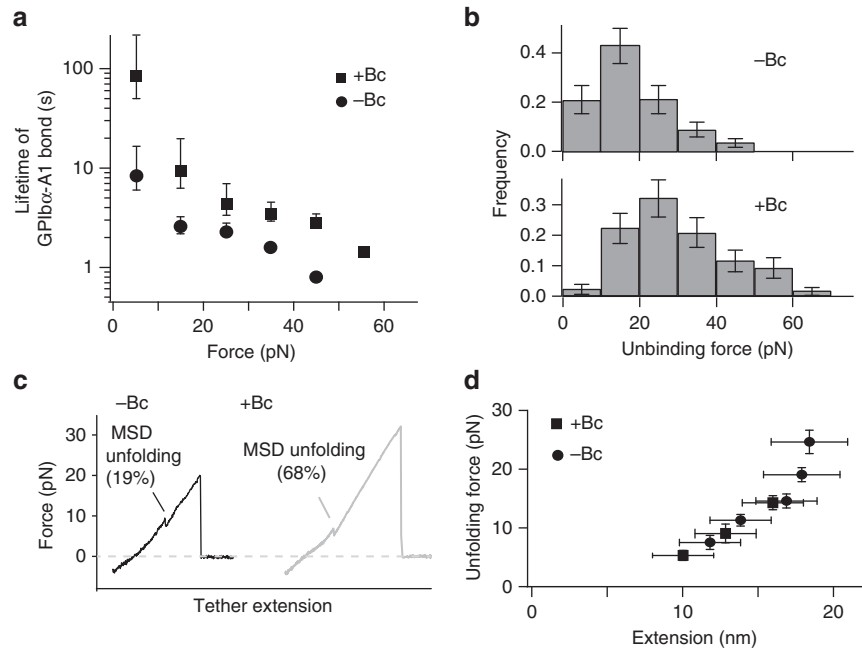

**Figure 2 | Botrocetin facilitates mechanical pulling-induced unfolding of MSD.** The A1-coated bead was placed in an optical trap and used to engage and pull on the immobilized GPIb–IX-coated bead in the experimental buffer that contained either no ( − Bc) or 1 μg ml⁻¹ botrocetin ( + Bc). (**a**) Plot of lifetimes of the GPIb–IX/A1 bond as a function of force. Error bars are Poisson noise[68]. (**b**) Histograms of the unbinding forces between GPIb–IX and A1 under a pulling rate of ∼150 nm s⁻¹. In each histogram, the Y axis is normalized by the total number of unbinding events ($n = 71$–90). Error bars are Poisson noise. (**c**) Representative force-distance traces of pulling A1 on GPIb–IX. For GPIb–IX, frequencies of observing MSD unfolding were 19% in the absence of botrocetin, and 68% in its presence. (**d**) Force-extension plots for the MSD unfolding in the absence and presence of botrocetin. Extension distances were sorted by unfolding force into 4-pN bins.

**GPIb–IX with unfolded MSD exhibits constitutive signalling.** We have previously identified a mutant GPIbα, designated GPIbαΔ, in which a significant portion of the MSD (residues 443–471) is removed, leaving the remaining MSD residues unfolded[14] (Fig. 4a,b). Nevertheless, GPIbαΔ assembly with GPIbβ and GPIX, and its interaction with A1 are wild type (WT)-like[14] (Supplementary Fig. 6). Here we tested whether GPIbαΔ, mimicking unfolded MSD, affects GPIb–IX signalling using the filopodia assay. In this assay, platelets or transfected cells expressing GPIb–IX are placed onto a VWF-coated surface in the presence of botrocetin and EDTA, the filopodia formation in the attached cell depends on the VWF/GPIb–IX engagement and is an effective indicator of GPIb–IX signalling[32–34]. Inducible expression of WT GPIbα and mutant GPIbαΔ, along with GPIbβ and GPIX, was engineered in transfected Chinese hamster ovary (CHO) cells such that, on doxycycline induction, GPIbα and GPIbαΔ were expressed at comparable levels (Fig. 4c). Both cells attached to the VWF surface in a botrocetin- and doxycycline-dependent manner (Fig. 4d). Confocal microscopic analysis of the attached cells revealed that, in accordance with earlier reports[32–34], filopodia in CHO cells expressing GPIbα formed mostly at the bottom in contact with VWF (Fig. 4e). In comparison, filopodia in CHO cells expressing mutant GPIbαΔ formed over the entire cell surface, including where there was no VWF (Fig. 4e,f). Adding anti-Ibβ monoclonal antibody RAM.1, which inhibits the filopodia formation in platelets or cells expressing WT GPIb–IX without affecting VWF binding to GPIb–IX[34,35], inhibited the filopodia formation in CHO cells expressing GPIbαΔ (Fig. 4g,h, Supplementary Fig. 7). Overall, these results suggest that GPIbαΔ could induce cellular signalling independent of VWF binding and that the signalling propagates through the nearby GPIbβ subunit.

**Shear and ligand induce clearance signals in mice.** Botrocetin's facilitation of the VWF/GPIb–IX interaction is not species-dependent[16], making it possible to assess the function of botrocetin-mediated GPIb–IX signalling in animals. Consistent with earlier reports that infusion of botrocetin induces rapid clearance of platelets and associated VWF in pigs, dogs and rats[36,37], intravenous injection of botrocetin (5 μg g⁻¹ of body weight) into WT C57BL/6J mice induced within an hour a precipitous 80% drop in platelet count, which gradually recovered in 3 days (Fig. 5a,b). In a separate experiment, citrated murine PRP obtained from C57BL/6J mice underwent the uniform shear treatment in the absence and presence of 2 μg ml⁻¹ botrocetin. Afterwards, the platelets were collected for either flow analysis as described above or clearance study with intravenous infusion into a recipient mouse (Fig. 5c). Botrocetin/shear, but not shear alone, induced the same GPIb–IX-mediated signals in murine platelets as those in human platelets (Fig. 5d; Supplementary Fig. 8). Moreover, all botrocetin/shear-treated platelets were cleared in mice within an hour of infusion (Fig. 5e). In contrast, shear-treated platelets were cleared gradually as endogenous ones. It is important to note that infusion of botrocetin itself into mice, at the amount used in the *in vitro* shear treatment (∼0.05 μg g⁻¹), could not induce any significant clearance of platelets (Supplementary Fig. 9). Overall, these results indicate that botrocetin/shear-mediated GPIb–IX signalling causes platelet clearance.

**Exposed trigger sequence in GPIbα induces platelet clearance.** When MSD becomes unfolded on shear-mediated mechanical pulling, residues in the MSD are expected to adopt an extended conformation, at ∼3–4 Å per residue[14]. According to a recent

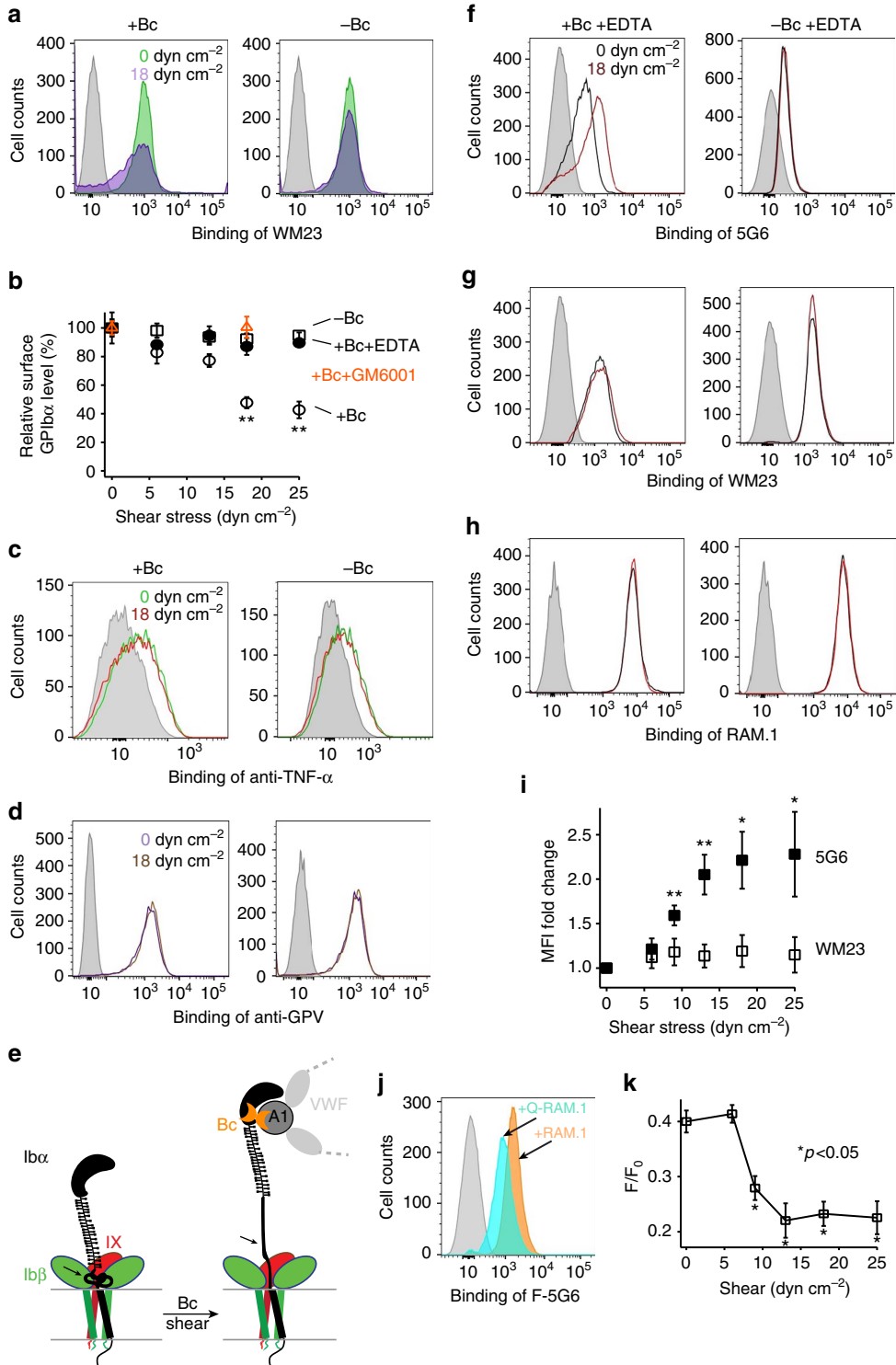

**Figure 3 | Botrocetin and physiological shear induce deformation of MSD on the platelet.** (**a**) Flow histograms illustrating that botrocetin/shear reduced GPIbα level in human platelets as measured by binding of WM23. Grey: IgG control; green: no shear; purple: 18 dyn cm$^{-2}$ shear. (**b**) Quantified relative GPIbα levels as a function of shear stress. Median fluorescence intensity (MFI) under each condition was normalized to that under no shear being 100%. Open squares: with botrocetin (1 μg ml$^{-1}$) and EDTA (+ Bc + EDTA); filled circles: without botrocetin (− Bc); open circles: with botrocetin (+ Bc); open orange triangles: with botrocetin and GM6001 (10 μM). Data are shown as mean ± s.d. (n = 3). P values are between + Bc and − Bc. **P < 0.01. (**c,d**) Flow histograms illustrating the lack of change in pro-TNFα and GPV levels following botrocetin/shear treatment. (**e**) A diagram illustrating botrocetin/shear induces unfolding of the MSD in GPIb–IX, adapted from[14]. The ADAM17 cleavage site in the MSD is marked by the arrowhead. (**f–h**) Flow histograms illustrating botrocetin/shear-induced increase in 5G6 binding, but not in WM23 or RAM.1 binding. EDTA was added during treatment to prevent GPIbα shedding. (**i**) Quantificative plots of change in 5G6 binding versus WM23 binding as a function of shear stress. Filled square: 5G6 binding; open square: WM23 binding. Data are shown as mean ± s.d. (n = 3). *P < 0.05. (**j**) Flow histograms illustrating fluorescence quenching of fluorescein-labelled 5G6 (F-5G6) by quencher-labelled RAM.1 (Q-RAM.1) on resting platelets in the absence of shear. (**k**) Shear-dependent change of the quenching efficiency (F/F$_0$), in which F is the MFI value obtained with F-5G6 and Q-RAM.1, and F$_0$ is that with F-5G6 and unlabelled RAM.1. Data are shown as mean ± s.d. (n = 3). P values are between shear and no shear treatment.

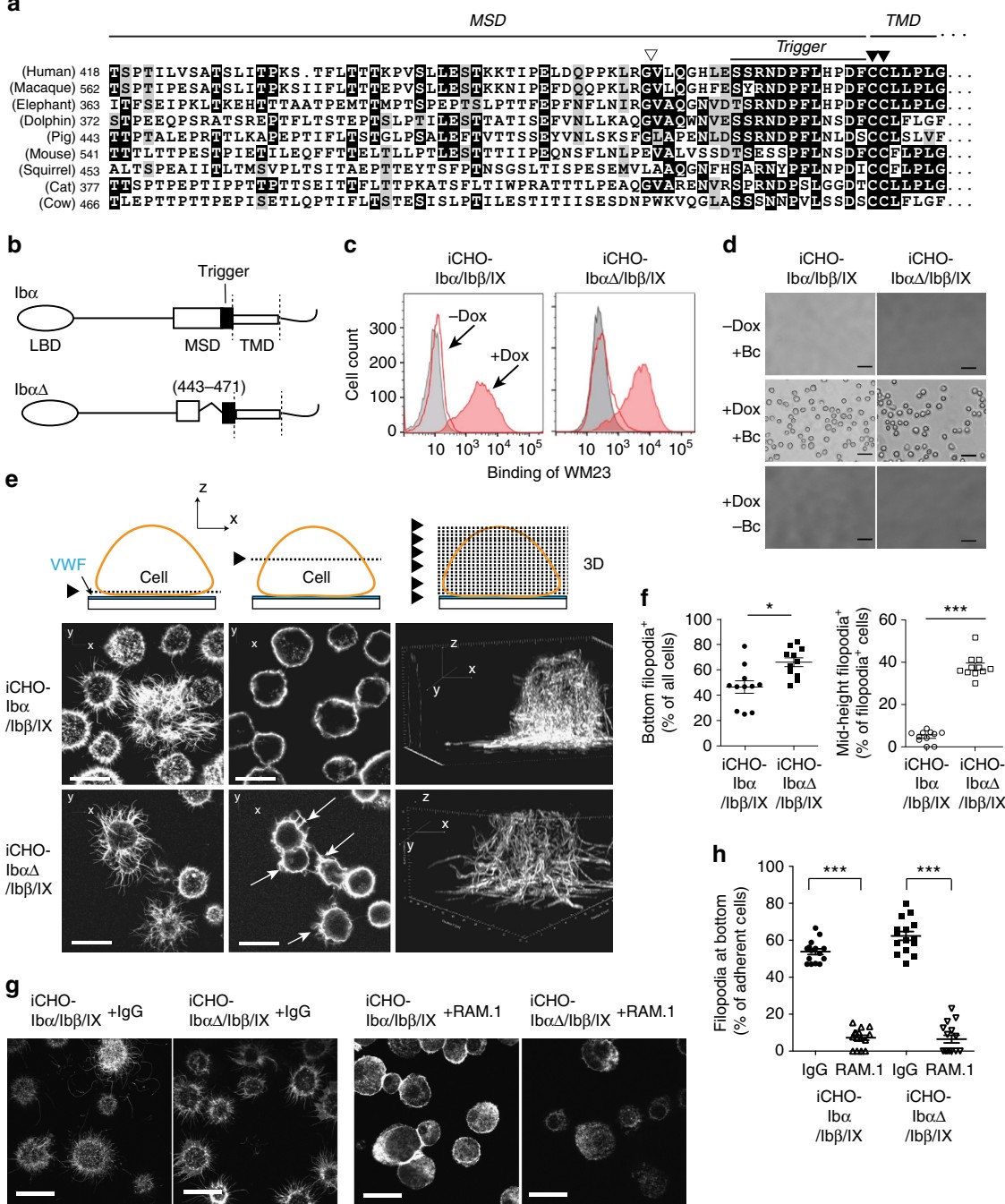

**Figure 4 | A deletion mutation unfolding the MSD causes ligand-free GPIb–IX signalling.** (**a**) Sequence alignment of the MSD from selected species. Sequences are obtained from the Ensembl Genome Database. The alignment is anchored at the conserved vicinal cysteine residues marked by filled triangles. Highly and weakly conserved residues are shaded black and grey with BOXSHADE program, respectively. The ADAM17 cleavage site is marked by an open triangle. (**b**) Schemes of MSD-unfolding mutant GPIbαΔ and wild-type GPIbα. In GPIbαΔ, residues 443–471 in the MSD were deleted. The Trigger sequence in MSD is coloured black. (**c**) Flow histograms of GPIbα on inducible CHO cells that expressed wild-type (iCHO–Ibα/Ibβ/IX) or mutant (iCHO–IbαΔ/Ibβ/IX) complex. Grey filled: untransfected cells; red open: transfected cells without doxycycline induction (−Dox); red filled: transfected cells after doxycycline induction (+Dox). (**d**) Microscopic images showing that transfected cells adhered to the VWF surface only on doxycycline induction and in the presence of botrocetin (Bc). Scale bar, 40 µm. (**e**) GPIbαΔ, but not wild type, induced ligand-free filopodia formation in transfected cells. Top row: cartoons of cell and VWF-coated slide, with triangles marking the *z* position of the viewing panel(s) at: (left) near bottom, in proximity with coated VWF; (middle) middle of the cell, where VWF was absent; and (right) three-dimensional reconstruction of a representative cell. Middle row: images of iCHO–Ibα/Ibβ/IX cells. Bottom row: iCHO–IbαΔ/Ibβ/IX cells; white arrowheads mark the filopodia extrusion in the absence of VWF. Scale bar, 10 µm. (**f**) Quantificative comparison of filopodia formation at the bottom (left) and in the middle (right) of adherent cells. Cells with >3 filopodia of >2-µm length were considered positive. A total of 200 cells from 11 view fields were visually examined and counted. Data are shown as mean ± s.d. from three independent experiments. *$P<0.05$; ***$P<0.001$. (**g**) Representative images (bottom view) showing that RAM.1, not rat IgG, inhibits the filopodia of iCHO–Ibα/Ibβ/IX and iCHO–IbαΔ/Ibβ/IX cells. (**h**) Quantificative comparison of filopodia of IgG- and RAM.1-treated cells. The cells from 15 view fields were visually examined and counted. Data are shown as mean ± s.d. from three independent experiments. ***$P<0.001$.

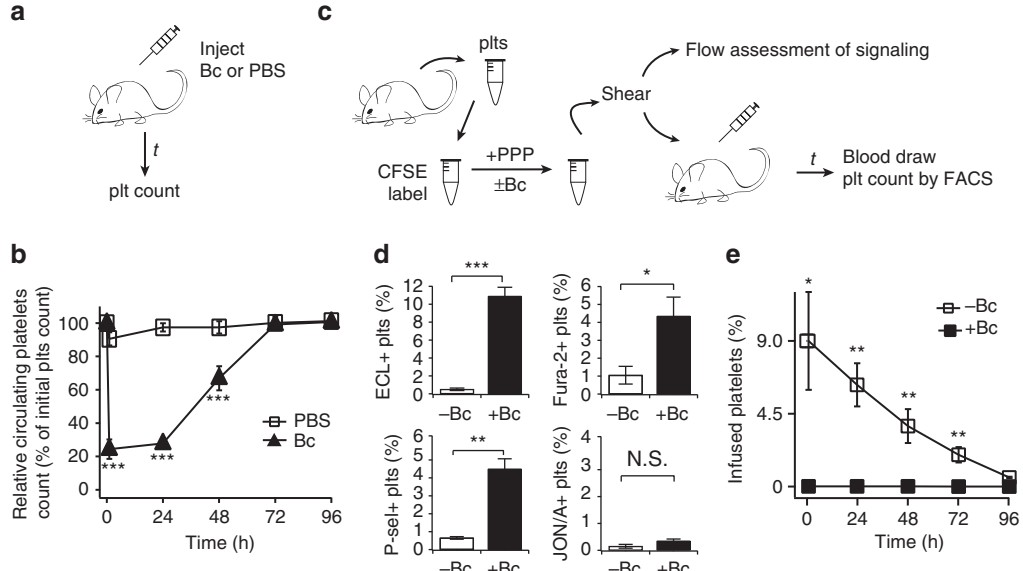

**Figure 5 | Botrocetin and shear cause GPIb–IX signalling and clearance of murine platelets. (a)** Illustration of the clearance study of botrocetin (Bc) injection. Purified botrocetin or PBS was injected intravenously into a C57BL/6J mouse, and blood was drawn at noted time points for platelet count. **(b)** Relative platelet count following botrocetin or PBS injection, with that before injection being 100%. Data are shown as mean ± s.d. ($n = 6$). **(c)** Illustration of the clearance study of *in vitro* sheared platelets. Murine platelets were collected from C57BL/6J mice, fluorescently labelled before being mixed with platelet-poor plasma (PPP) and botrocetin. The reconstituted PRP was treated with uniform shear of 18 dyn cm$^{-2}$ before being either analysed by flow cytometry or being infused into a recipient mouse for clearance measurement. The PE-conjugated JON/A antibody selectively binds to the activated murine integrin αIIbβ3. **(d)** Botrocetin/shear treatment induces similar signalling in murine platelets. Quantificative plots of platelet signalling are plotted as described in Fig. 1b. **(e)** Clearance of *in vitro* sheared murine platelets, expressed as the percentage of CSFE-labelled platelets in total platelet population, after infusion. Data are shown as mean ± s.d. ($n = 6$). *$P < 0.05$; **$P < 0.01$; ***$P < 0.001$.

model of GPIb–IX[8,38], the height of GPIbβ/GPIX extracellular domains is ~30 Å. Thus, on unfolding MSD residues in direct contact with GPIbβ/GPIX are likely to be ~10 residues immediately preceding the transmembrane domain (that is, residues 473–483, termed Trigger), which are retained in GPIbαΔ (Fig. 4a). Since GPIbαΔ can induce signalling without ligand binding and shear pulling, it is conceivable that the Trigger sequence in an unfolded and extended conformation is sufficient to trigger GPIb–IX signalling and platelet clearance. Consistently, whereas most residues in the MSD are not conserved across species, many residues in the Trigger sequence are (Fig. 4a).

In an earlier study, a chimeric protein called interleukin-4 receptor (IL4R)-Ibα, in which the extracellular domain of human GPIbα (residues 1–472) was replaced with that of the α-subunit of IL4R, was constructed[39]. IL4R-Ibα, like GPIbαΔ, contains the Trigger sequence but not the rest of MSD (Fig. 6a). In the absence of folded MSD, the Trigger sequence should be unfolded on the IL4R-Ibα transgenic (IL4R-IbαTg) platelet. To ascertain whether there is constitutive GPIb–IX signalling in IL4R-IbαTg platelets, washed platelets were obtained from whole blood of WT C57BL/6J and IL4R-IbαTg mice, and analysed without botrocetin/shear treatment for aforementioned indicators of GPIb–IX signalling. Microscopic images of these platelets revealed that ~40% of IL4R-IbαTg platelets exhibited filopodia in the absence of bound GPIbα ligand, markedly higher than <10% positive rate for WT ones (Fig. 6b,c). Similarly, compared with WT, IL4R-IbαTg platelets displayed significantly higher intracellular calcium concentration, and small but reproducibly higher surface P-selectin expression level (Fig. 6d; Supplementary Fig. 10a). IL4R-IbαTg platelets exhibited similar ECL binding level to the WT (Fig. 6d). Since the replacement of the heavily glycosylated extracellular domain of GPIbα with that of IL4R would likely result in a reduction of the overall glycosylation level on the IL4R-IbαTg platelet, it may be difficult to interpret the ECL binding

level and correlate it solely with the exposure of β-galactose. Overall, because IL4R-Ibα cannot bind VWF or other GPIbα ligands, these results support the presence of constitutive GPIb–IX signalling in IL4R-IbαTg platelets. Finally, IL4R-IbαTg mice have a significantly lower platelet count than WT mice[39] (Supplementary Fig. 10b). On infusion, a significant portion of IL4R-IbαTg platelets (~40%), but not WT ones, were cleared within an hour of infusion (Fig. 6e,f). IL4R-IbαTg platelets were cleared at a faster rate than WT (Fig. 6g,h). Overall, these results suggest that the unfolded Trigger sequence on the surface of IL4R-IbαTg platelets induces ligand-free GPIb–IX signalling and platelet clearance.

## Discussion

It has been recognized for decades that shear flow or stirring is required for initiating GPIb–IX-mediated signalling and activation in platelets, but the molecular mechanism underlying the shear requirement has remained unclear[1]. In this study, we have illustrated the initial molecular event platelets undertake in response to shear stress, and provided the evidence demonstrating the juxtamembrane domain in GPIbα as a MSD. Binding of VWF under physiological shear induced MSD-unfolding and intracellular signalling events in the platelet. In addition, mutations that unfolded the MSD, and the juxtamembrane Trigger sequence therein, induced ligand-free GPIb–IX signalling and platelet clearance. On the basis of these results, we propose a 'trigger' model of GPIb–IX signalling (Fig. 7) that can explain the shear requirement; in the resting platelet, the MSD including the Trigger sequence is folded. Ligand binding to the LBD under shear stress exerts a pulling force on GPIb–IX and induces unfolding of MSD. Consequently, the Trigger sequence becomes extended and presumably exposed to nearby GPIbβ and GPIX extracellular domains, setting off

GPIb–IX signalling into the cell. Downstream signals include the increase of intracellular calcium, increased surface expression of P-selectin, filopodia formation and glycan changes.

Thrombocytopenia is a common symptom in type 2B VWD patients[22,40], which was recapitulated recently in a VWF transgenic murine model[41]. In these type 2B VWD mice,

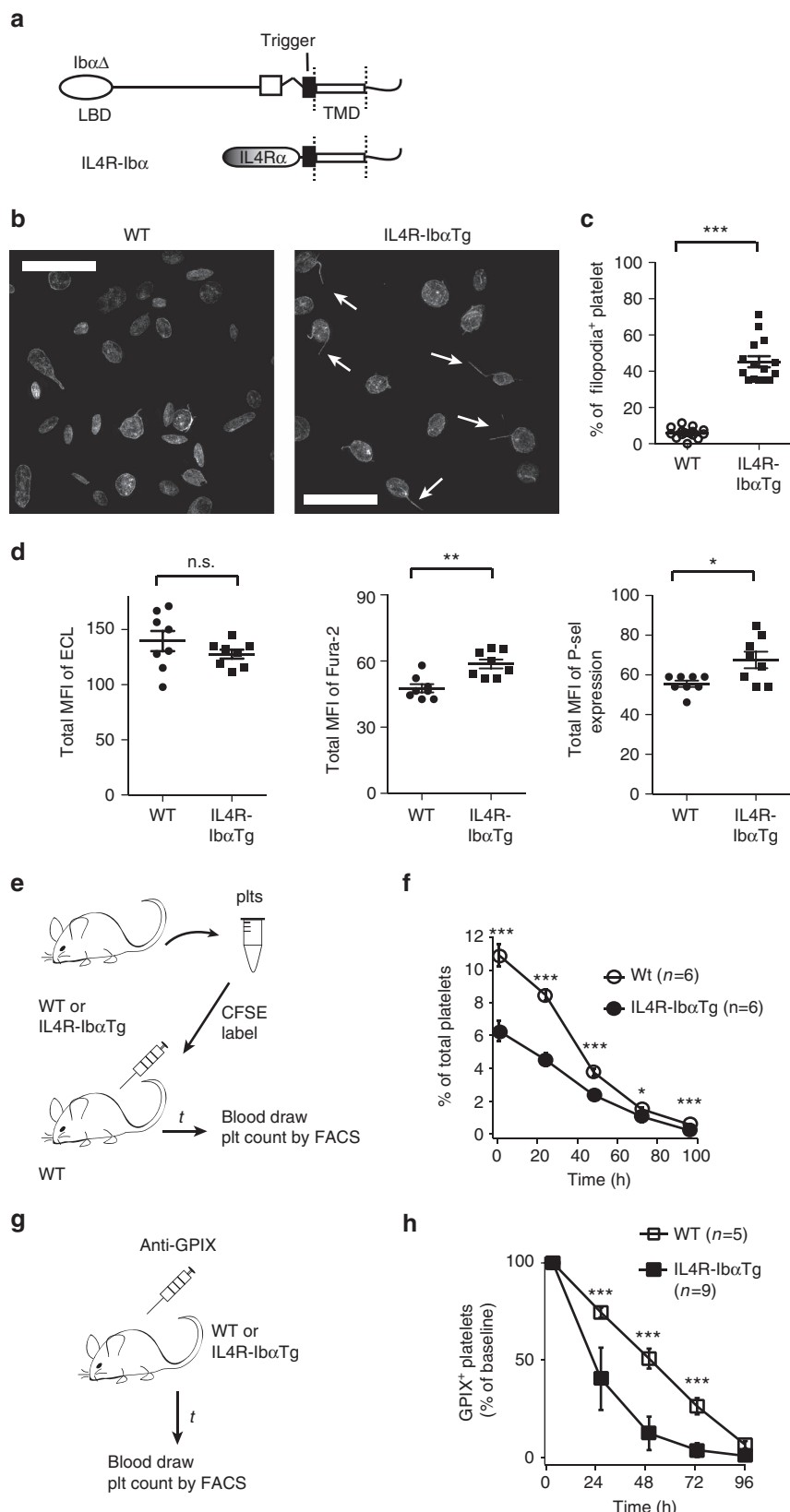

macrophages are involved in clearing the VWF-bound platelets[41]. Similarly, thrombocytopenia is quickly induced in animals on injection of botrocetin[36,37] (Fig. 5), but the underlying molecular mechanism has remained unclear. In this study, we show for the first time that botrocetin/VWF or type 2B VWF under physiological shear can induce exposure of β-galactose on the platelet (Fig. 1). The exposed β-galactose has been suggested to mediate platelet clearance during sepsis or after cold storage through its interaction with the Ashwell–Morell receptor[13,42]. Relatedly, cold storage induces surface expression of neuraminidases on the platelet surface[43]. It was reported recently that many anti-LBD antibodies that cause Fc-independent platelet clearance in mice induce surface

expression of neuraminidases and exposure of β-galactose[44]. In a separate study of anti-LBD antibodies, the exposure of N-acetyl-glucosamine was also implicated to mediate platelet clearance by macrophages[45]. Since macrophages can uptake platelets displaying altered glycans[45,46], it is conceivable that MSD-unfolding-induced alteration of platelet glycans may help to mediate fast clearance of platelets in type 2B VWD patients. It is not clear how GPIb–IX signalling leads to the exposure of β-galactose or other glycan changes. One possibility is that GPIb–IX signalling leads to granule release, through which neuraminidases are translocated from the lysosome to the plasma membrane. Consistently, more P-selectin was detected on the platelet surface following botrocetin/VWF or type 2B VWF treatment under shear (Figs 1 and 5). It is noteworthy that the extent of P-selectin expression here was significantly smaller than that induced by thrombin activation, suggesting that GPIb–IX-induced granule release is of limited scale.

A critical feature of the 'trigger' model is that a pulling force, rather than a conformational change in the LBD, is transmitted through the long macroglycopeptide region (Fig. 7). Thus, whether the bound ligand can sustain the pulling to efficiently induce MSD unfolding (for example, unbinding force of the LBD/ligand complex > unfolding force of MSD), instead of inducing LBD to adopt a specific conformation[47,48], may determine the onset of GPIb–IX signalling. In this study, we demonstrated in the optical tweezer experiment that botrocetin increased the unbinding force between A1 and GPIb–IX (Fig. 2). Consistently, it takes much longer time for a rolling platelet under fluid shear to detach from immobilized A1 domain bearing the V1316M mutation than from the WT A1 (ref. 49). On binding, botrocetin/VWF and VWF.V1316M may induce different conformations of LBD, both increased the unbinding force, induced the same signals in platelets and induced platelet clearance (Figs 1 and 5). Likewise, many monoclonal anti-LBD antibodies do not share a common epitope and they probably do not bind LBD as VWF does, they nonetheless are capable of inducing GPIb–IX signalling and platelet clearance in an Fc-independent manner[12,44,45,50–52]. Considering the very similar effects induced by botrocetin/VWF and anti-LBD antibodies[44], including the time course of platelet clearance following injection (Fig. 5b), it is tempting to speculate that, like botrocetin/VWF, these anti-LBD antibodies bind LBD with unbinding forces that are sufficient to induce MSD unfolding.

CHO cells expressing GPIbαΔ and IL4R-IbαTg platelets, both of which contain a constitutively extended Trigger sequence in their respective mutant GPIb–IX complexes, exhibited GPIb–IX signalling in the absence of bound ligands (Figs 4 and 6). Without ligand binding, no pulling force is exerted on GPIb–IX in these cells, yet similar GPIb–IX signalling, in terms of filopodia formation, intracellular calcium concentration and P-selectin expression, as well as accelerated clearance in vivo, was observed for these cells compared with botrocetin/shear- and type 2B

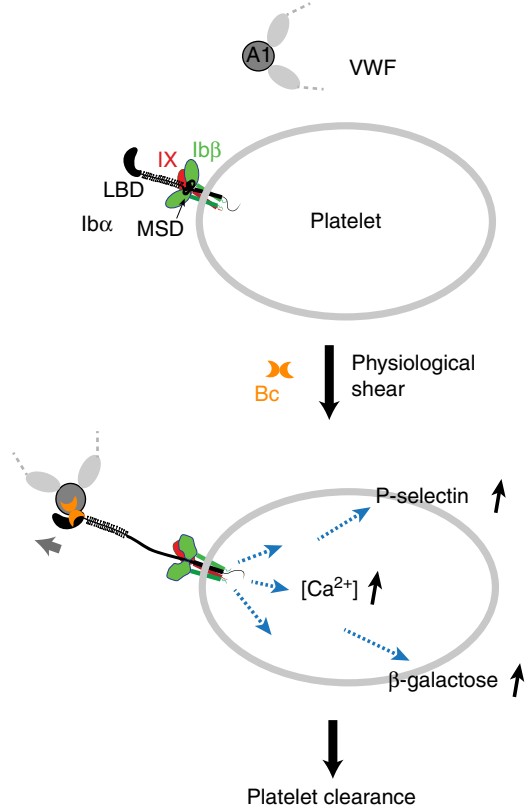

**Figure 7 | The trigger model of GPIb–IX signalling and platelet clearance.** In the resting platelet (top), the MSD in GPIb–IX is folded. Plasma VWF does not interact with GPIb–IX on the platelet. In the presence of botrocetin (Bc) and physiological shear, VWF binds to the LBD in GPIb–IX and pulls on the complex to cause unfolding of MSD and the Trigger sequence therein. Consequently, it induces increase in the intracellular calcium level, expression of P-selectin and exposure of β-galactose on the platelet surface, leading to rapid clearance of platelets.

**Figure 6 | IL4R-IbαTg platelets exhibit ligand-free GPIb–IX signalling and fast clearance.** (**a**) Schemes of MSD-unfolding mutant GPIbαΔ and IL4R-Ibα. Both proteins contain the Trigger sequence without the adjoining MSD residues. (**b**) Confocal fluorescence images of fixed washed platelets from C57BL/6J (WT, left) and IL4R-IbαTg mice (right). White arrowheads mark the filopodia extrusions. Scale bar, 10 μm. (**c**) Quantificative comparison of filopodia observed in the platelets. Platelets from 15 view fields (∼80–110 platelets per view field) were visually examined and counted. (**d**) Quantificative comparison of the exposure of β-galactose (measured by binding of FITC-labelled ECL), intracellular calcium level (monitored by Fura-2 fluorescence) and expression of P-selectin (binding of anti-P-selectin antibody) of the fixed washed platelets obtained from sex- and age-matched mice (n = 8 in each group). Data were quantified from the median fluorescence intensity of all the platelets (using the same gating as in Supplementary Fig. 1e). (**e**) Illustration of the clearance study of WT and IL4R-IbαTg platelets in WT mice. (**f**) Clearance traces of WT and IL4R-IbαTg platelets, expressed as the percentage of CSFE-labelled platelets in total population. (**g**) Illustration of the survival study of WT and IL4R-IbαTg platelets. Circulating platelets were labelled with fluorophore-labelled anti-GPIX antibody. (**h**) Clearance of these labelled platelets was monitored over time. Data are shown as mean ± s.d. *P < 0.05; **P < 0.01; ***P < 0.001; NS, no statistical difference.

VWF/shear-treated ones. It is noteworthy that the extent of GPIb–IX signalling in IL4R-IbαTg platelets appeared lower than that in botrocetin/shear-treated ones, which is consistent with the extent of clearance *in vivo* (Figs 1 and 6). The reason for the difference is not clear. One possibility is that the signals in botrocetin/shear-treated platelets were synchronized and thus appeared larger at the time of measurement. Another possibility is that botrocetin-associated VWF or type 2B VWF can induce additional effects such as the direct uptake by macrophages[41]. Overall, our results suggest that the extension or unfolding of the Trigger sequence may be the key step in setting off GPIb–IX signalling, even without a pulling force involved. How the Trigger sequence induces GPIb–IX signalling remains to be elucidated. One possibility is that the extended and exposed Trigger sequence initiates signalling by making contact with the nearby GPIbβ extracellular domains[8,53], as RAM.1 could inhibit constitutive filopodia formation in CHO cells expressing GPIbαΔ (Fig. 4g,h). These results also suggest that the extension of the Trigger sequence may be achieved not only by ligand/shear-induced unfolding of the MSD but also by proteolytic cleavage of the MSD. Shedding of GPIbα is a physiological process that occurs continuously on the surface of circulating platelets, releases the extracellular domain of GPIbα also known as glycocalicin into the plasma, and is largely mediated by ADAM17 (refs 25,54). It also occurs during storage of platelets in blood banks[55–57]. A tight correlation between GPIbα shedding and the severity of platelet storage lesion, particularly the post-transfusion survival of stored platelets, is well documented[58,59]. Consistently, specific inhibition of GPIbα shedding during platelet storage by exogenous inhibitors significantly reduced the fast clearance of senescent platelets in transfused mice[56–61], thereby establishing a causal–effect relationship between GPIbα shedding and platelet clearance. However, the underlying molecular mechanism is not clear. The ADAM17 cleavage site in GPIbα is at the Gly464–Val465 peptide bond[24], a few residues N terminal to the Trigger sequence (Fig. 4a). Thus, it is conceivable that GPIbα shedding leaves the remnant of the GPIbα extracellular domain, which includes the Trigger sequence, exposed and extended on the platelet surface in a similar manner as that on the IL4R-IbαTg platelet, thereby inducing GPIb–IX signalling and leading to platelet clearance.

Under physiological flow conditions, VWF in the plasma does not bind the platelet, with its A1 domain somehow shielded from binding to GPIbα (ref. 62). Under elevated shear stress, VWF undergoes a conformational change and assumes a high-affinity state for GPIbα (ref. 6). Thus, it is thought that the ligand/receptor pair of VWF/GPIbα responds primarily to elevated shear stress and is particularly useful in mediating platelet adhesion and thrombus formation under those conditions, in which the GPIbα association with filamin in the cytoplasm also plays a role[63]. In this study, we demonstrated that physiological shear stress is sufficient to induce MSD unfolding and platelet signalling through GPIb–IX. Elevated or complex shear conditions were avoided in this study and only shear stress within the physiological range was applied (Figs 1 and 3). Moreover, botrocetin instead of ristocetin was used to induce VWF binding of GPIbα, since previous studies suggested that, compared with botrocetin, ristocetin-induced binding mimics more closely the VWF/GPIbα interaction under elevated shear conditions and it induces additional effects in the platelet through GPIb–IX and other receptors[33,64]. In the case of CHO cell adhesion to immobilized VWF (Fig. 4), the adhered cell may generate contractile force through the cytoskeleton and exerts it on the ligated VWF/GPIbα pair[65]. The strength of such contractile force imposed on a single VWF/GPIbα pair is not clear, but it may conceivably be sufficient to induce MSD unfolding on the surface

of adhered cells. Our results suggest that with separate MSDs VWF and GPIb–IX may respond to shear via distinct mechanisms. Under circumstances in which VWF binds GPIbα in normal blood flow, such as type 2B VWD[22], VWF binding may induce MSD unfolding and GPIb–IX-transduced signalling in the platelet and lead to thrombocytopenia as discussed above. Other mechanosensory elements in VWF might not participate in this process, because CHO cells expressing mutant GPIbαΔ and IL4R-IbαTg platelets can signal in the absence of ligand/shear (Figs 4 and 6). It remains to be determined whether and how GPIb–IX responds to ligand binding under elevated shear stress, whether or how it responds act in concert with ligands and intracellular signalling molecules, particularly the mechanosensory elements therein. Overall, defining a role of GPIb–IX in inducing platelet signalling under physiological shear provides a foundation for future investigations of interplays between platelets, plasma and vessels under diverse shear flow conditions.

Many cell adhesion receptors are known force sensors. They often contain a LBD that is located distal to the cell membrane and linked to the transmembrane domain via a long repeating sequence and/or a heavily glycosylated region. In a Notch receptor, distal ligand binding and pulling induces unfolding of the juxtamembrane negative regulatory region, leading to exposure and proteolysis of the shedding cleavage site[66]. Shedding of the extracellular domain of Notch is a necessary step in Notch signalling, which proceeds with another cleavage of its transmembrane domain and relocation of its intracellular domain into the nucleus[67]. Although GPIbα bears little resemblance to Notch in sequence, structure or function, the 'trigger' model of GPIb–IX signalling is remarkably similar in two key aspects. First, the signal of ligand binding is transmitted as a mechanical force through a polypeptide sequence over a long distance. Second, the pulling force induces unfolding of a juxtamembrane MSD that effectively transduces the force information into a conformational change. It therefore seems reasonable to propose that receptor mechanosensing and unfolding may be an evolutionarily conserved and fundamental signalling mechanism used by cells to transmit information across the cell membrane.

## Methods

**Human subjects.** Citrated whole blood was drawn from healthy volunteers according to an approved protocol, in which all volunteers gave written informed consent. The collected whole blood was used to prepare PRP and plasma. Experiments involving fresh human platelets were performed in accordance with experimental protocols approved by the Institutional Review Board of Emory University (IRB#00006228). Plasma of a type 2B VWD patient was obtained in accordance with established protocols approved by the Institutional Review Board of University of Colorado Denver Anschutz Medical Campus (IRB#09-0816).

**Mice.** C57BL/6J mice were purchased from Jackson Laboratories. IL4R-IbαTg mice on the C57BL/6J genetic background have been described[39]. Six- to eight-week-old mice were used in all experiments except those involving IL4R-IbαTg. Both sexes of age-matched littermates were generally used for the study. We did not involve statistical analyses in which a pre-specified effect size was used, and were generally blinded to the group allocation. Siblings were randomly and alternatively selected for different treatments. The number of animals used in each experiment was indicated in each figure, given that phenotypes were reproducible. No specific inclusion/exclusion criteria were applied from the analysis. All experiments involving mice were performed in accordance with the protocols approved by the Institutional Animal Care and Use Committees (IACUC) of Emory University and University of North Carolina.

**Materials.** DMEM, L-glutamine, penicillin/streptomycin and non-essential amino acids were purchased from Mediatech (Manassas, VA, USA). Antibiotic G418 and lipofectamine 2000 were purchased from Life Technologies (Grand Island, NY, USA). Fetal bovine serum was purchased from Hyclone, Logan, UT, USA, hybridoma cloning factor from PAA, Etobicoke, Canada and hypoxanthine supplement from Sigma-Aldrich, St Louis, MO, USA. Human VWF, free of factor VIII, was

purchased from Haematologic Technologies, Inc (Essex Junction, VT, USA). Puromycin, doxycycline, snake venom from *Bothrops jararaca* and monoclonal anti-VWF antibody 1A11 (cat# SAB1402960) were from Sigma-Aldrich. Monoclonal antibody 5G6 and RAM.1 have been described before[28,31]. GM6001 was from Millipore (Billerica, MA, USA). Fluorescently conjugated ECL (cat# E3453-19C) was purchased from USBiological (Swampscott, MA, USA); phycoerythrin (PE)-labelled anti-P-selectin antibody (cat# 304906) and PAC-1 antibody (cat# 362802) were purchased from Biolegend (San Diego, CA, USA). QSY7 carboxylic acid succinimidyl ester (QSY7-NHS) and 5/6-carboxyfluorescein succinimidyl ester (CFSE) were purchased from Invitrogen (Carlsbad, CA, USA). WT and mutant GPIb–IX complexes in which the cytoplasmic domain of GPIX was biotinylated have been described[14].

**Purification of botrocetin.** Botrocetin was purified from *Bothrops jararaca* venom (Sigma) largely as described[23]. In brief, the lyophilized venom was dissolved in 0.01 M Tris–HCl, 0.15 M NaCl, pH 7.4 (TS buffer), and fractionated by 60–80% ammonium sulfate precipitation at 22 °C. The precipitate was dissolved in TS buffer and dialysed against the same buffer at 4 °C overnight. After elution from a diethylaminoethyl cellulose column via a linear gradient of 0.15–0.4 M NaCl, the botrocetin-containing fractions were concentrated and further purified by gel filtration chromatography using a Superdex 200 column pre-equilibrated with TS buffer. The botrocetin activity was assayed for its ability to induce VWF binding to immobilized human GPIb–IX[35]. Fractions with peak activities were concentrated using an Amicon ultracentrifugal filter with ultracel-3 membrane and stored at − 80 °C. The concentration was measured by the absorbance using an extinction coefficient of 2.985 ml mg$^{-1}$ cm$^{-1}$.

**Uniform shear assay.** Freshly prepared human or murine PRP, supplemented with plasma to a count of $0.2$–$5 \times 10^5$ platelets per μl, was mixed gently with 1 μg ml$^{-1}$ botrocetin or other noted additives, incubated at room temperature for 10 min, and transferred to the stationary plate surface of a CAP2000 + cone-plate viscometer (Brookfield Engineering Laboratories, Middleboro, MA, USA). Primary shear rate varied from 0 to 25 dyn cm$^{-2}$ (0–2,533 s$^{-1}$). After shear treatment of 1–2 min at room temperature, ∼50 μl of PRP mixture was collected gently, treated with desired monoclonal antibodies or noted detecting agents for 10 min, washed with modified Tyrode's buffer when desired, and fixed by the addition of 200 μl 4% paraformaldehyde (PFA). For the measurement of MSD extension, 5G6 and other noted antibodies were added to the PRP mixture at 0.5 μg ml$^{-1}$ concentration before shear treatment. After shear, the mixture was collected and immediately fixed with 4% PFA.

**Flow cytometry analysis.** Fixed platelets were analysed on a BD FACS Canto II flow cytometer using FlowJo software. Median fluorescence intensity of each cell population (10,000 cells) was obtained for quantification and comparison. The forward-/side-scattered light (FSC/SSC) voltages were set at 199/400, respectively. The same gating (shown in Supplementary Fig. 1e) was applied to all the platelet samples analysed in this study. For plasma VWF binding, platelets in PRP were incubated with a rabbit anti-VWF polyclonal antibody and 1 μg ml$^{-1}$ botrocetin for 10 min at 20 °C, then fixed with 4% PFA before flow analysis. As a negative control, the platelet in PRP was incubated with secondary antibody and analysed in parallel. For the measurement of intracellular calcium influx, the sample treated with only solvent was used as the negative control.

**Laser optical tweezer measurement.** Single-molecule force measurement was performed as described[14]. In brief, streptavidin-coated beads were incubated for 10 min with 1 nM biotin–DNA handle nitrilotriacetic acid in Tris-buffered saline (150 mM NaCl, 10 mM Tris–HCl, 5 mM NiCl$_2$, pH 7.5). The beads were washed and incubated with 100 pM recombinant hexahistidine-tagged A1 domain (VWF residues Asp1261–Pro1466) for 15 min before the experiment. Recombinant GPIb–IX complex in which the GPIX cytoplasmic tail was biotinylated was coupled with the streptavidin-coated beads by incubating the beads with 20 μl cell lysate containing biotinylated GPIb–IX for 10 min and washed with Tris-buffered saline containing 1% Triton X-100. The single-molecule pulling experiments were performed using an analytical minioptical tweezer apparatus[14,68] in the presence and absence of 1 μg ml$^{-1}$ botrocetin. Force and bead-to-bead distance were recorded at 200 Hz. When appropriate, the force-extension data were fitted to the worm-like chain model. The lifetime of bond as a function of force was estimated using the Dudko–Hummer–Szabo equation.

**Preparation of conjugated antibodies.** Purified 5G6 and RAM.1 were conjugated with CFSE and QSY7-NHS, respectively, following the manufacturer's instruction. In brief, 1 mg antibody in the reaction buffer (2 mg ml$^{-1}$ in 0.1 M sodium phosphate, 150 mM NaCl, pH 7.4) was added to ∼10 μg of reactive dye dissolved in anhydrous dimethylsulphoxide in a glass vial, and incubated in the dark for 1 h at room temperature. After the reaction, unconjugated dye was separated from labelled antibody on a PD-10 desalting column (GE Healthcare). Labelled antibodies were stored at − 20 °C, and their concentrations estimated using a Pierce BCA protein assay kit (Life Technologies).

**Fluorescence resonance energy transfer measurement.** Fluorescein-conjugated 5G6 (F-5G6) was mixed with either unconjugated RAM.1 or QSY7-conjugated RAM.1 (Q-RAM.1) at 1:5 mass ratio, and then added to the PRP mixture containing 1 μg ml$^{-1}$ botrocetin. The final 5G6 concentration was 0.5 μg ml$^{-1}$. Each mixture underwent uniform shear treatment as described above before being analysed by flow cytometry. The median fluorescence intensity value of each sample is considered as $F$, with that of platelet treated with F-5G6 and unlabelled RAM.1 as $F_0$.

**Construction of inducible CHO cells expressing GPIb–IX.** The CHO K1 cell line was purchased from American Type Culture Collection (Manassas, VA, USA, cat# CCL-61). The Tet-on 3G inducible expression system was purchased from Clontech (Mountain View, CA, USA) and stable CHO cell lines were established following the manufacturer's instruction. In brief, the Tet3G transactivator was transfected into CHO cells using Lipofectamine 2000. Stable clones underwent selection in culture media containing 500 μg ml$^{-1}$ G418 and maintained in that with 100 μg ml$^{-1}$. Individual CHO/Tet3G clones were screened for top induction using a firefly luciferase reporter under the Tet-inducible promoter in the presence and absence of 2 μg ml$^{-1}$ doxycycline. The CHO/Tet3G$^+$ cells were transfected with expression vectors in which transcription of GPIb–IX genes was under the control of Tet-inducible promoter[14]. Positive stable cells were sorted repeatedly for positive surface expression of GPIb–IX on induction of 2 μg ml$^{-1}$ doxycycline[14]. To verify the induced expression level of GPIb–IX, sorted cells were amplified, induced with doxycycline, collected (100,000 cells per 100 μl) and incubated with 0.5 μg ml$^{-1}$ monoclonal antibody WM23 in cold PBS. The stained cells were washed, incubated with allophycocyanin (APC)-labelled goat anti-mouse antibody and measured by flow cytometry using a BD Canto II FACS instrument[8,69].

**Fluorescence microscopy of filopodia formation.** For CHO cells, glass slide was coated with human VWF at 10 μg ml$^{-1}$ in PBS at 4 °C overnight and blocked with 1% bovine serum albumin in PBS for 1 h at 22 °C. CHO cells were induced for GPIb–IX expression in culture media containing 2 μg ml$^{-1}$ doxycycline for 1 day. The cells were then pelleted and resuspended at $1 \times 10^6$ cells per ml in modified Tyrode's buffer (134 mM NaCl, 0.34 mM Na$_2$HPO$_4$, 2.9 mM KCl, 1 mM MgCl$_2$, 5 mM glucose, 12 mM NaHCO$_3$, 20 mM HEPES, pH 7.35) containing 5 mM EDTA. Adhesion of CHO cells to the VWF-coated glass slide was performed largely as described[34]. In brief, CHO cells were placed on VWF-coated slides in the presence or absence of 1 μg ml$^{-1}$ botrocetin for 30 min at 37 °C. The adherent cells on the slide were washed with PBS buffer, fixed with 4% PFA for 10 min, permeabilized with 0.1% Triton X-100 for 15 min and stained with 2 μg ml$^{-1}$ TRITC-conjugated phalloidin for 30 min. For platelets, fresh washed platelets were prepared in modified Tyrode's buffer to a platelet count of $5 \times 10^6$ ml$^{-1}$. The platelets were gently mixed with equal volume of 4% PFA for 30 min at 37 °C. The fixed samples were applied onto an uncoated glass surface and incubated for another 30 min at 37 °C. The liquid buffer was carefully replaced by staining buffer containing 2 μg ml$^{-1}$ TRITC-conjugated phalloidin and 0.1% Triton X-100 in PBS via a vacuum pump and stained for 30 min. Cell images were acquired on a super-resolution DeltaVision OMX imaging system (GE Healthcare). Z-stack imaging was performed at 0.125 μm per step and three-dimensional reconstruction using IMARIS software (Bitplane).

**Botrocetin-induced platelet clearance in mice.** A single dose of botrocetin dissolved in PBS or PBS was injected intravenously into 6-week-old C57BL/6J mice. Before injection, 1 h after injection and every 24 h thereafter, a small blood sample was collected from the animal and platelet count was measured using a CBC counter.

**Clearance of in vitro sheared platelets in mice.** C57BL/6J mice, under isoflurane anaesthesia, were bled from the retro-orbital plexus into sodium citrate buffer. PRP was obtained by centrifugation at 300*g* for 10 min, and platelets isolated at 1,000*g* for 5 min. Platelet-poor plasma was kept. Platelets were stained with 2 μg ml$^{-1}$ CFSE in Tyrode's buffer containing 0.02 U ml$^{-1}$ apyrase and 0.1 μg ml$^{-1}$ PGI$_2$ for 30 min. After staining, platelets were washed once, suspended in platelet-poor plasma with or without 1 μg ml$^{-1}$ botrocetin to ∼$5 \times 10^5$ μl$^{-1}$, and underwent the treatment of uniform shear (18 dyn cm$^{-2}$) as described above. After shear, platelets were collected and analysed for signalling indicators by flow cytometry as described above. Instead of PAC-1, antibody JON/A was used to detect the activation of mouse αIIbβ3. Alternatively, after shear platelets were collected and directly infused into recipient C57BL/6J mice (∼$1 \times 10^8$ platelets per mouse). After 1 h and every 24 h thereafter, a small blood sample was collected from recipient mice, labelled with anti-CD41 antibody and analysed by flow cytometry. The percentage of infused platelets was calculated as the ratio of CFSE$^+$CD41$^+$ platelets versus CFSE$^-$CD41$^+$ platelets.

**Clearance of IL4R-IbαTg platelets.** Six- to forty-week-old male mice were used in this study. Platelets from C57BL/6J or IL4R-IbαTg mice were isolated in modified Tyrode's buffer, stained with CFSE as described above, washed once and pooled to $1.5 \times 10^6$ μl$^{-1}$. Platelets were infused right away into recipient C57BL/6J mice

($1.5 \times 10^8$ platelets per 10 g body weight). After 1 h and every 24 h thereafter, blood was drawn from recipient mice and infused platelets were counted as described above. To measure endogenous survival of IL4R-IbαTg platelets, a single intravenous injection of 5 μg AlexaFluor 488-conjugated anti-GPIX antibody (clone Xia.B4, Emfret analytics) in 100 μl PBS was administered at $t = 0$. Whole blood was drawn after every 24 h, diluted and incubated with a PE-conjugated anti-CD41 antibody (MWReg30, BD Biosciences) for 10 min at room temperature. The ratio of AlexaFluor 488-positive platelets to PE-positive ones was determined.

**Statistical analysis.** Data are expressed as mean ± s.d. An unpaired or paired two-tailed Student's *t*-test analysis was performed for statistical analyses. Sample size ranged from 3 to 9 as indicated. Differences were considered statistically significant when $P < 0.05$. The Mann–Whitney test was performed to test the variation similarity between groups that are being statistically compared and all the tested groups showed similar variations. All analyses were performed using GraphPad Prism software (version 6.0).

**Data availability.** The data that support the findings of this study are available in the manuscript and from the corresponding author on request.

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

## Acknowledgements

This paper is dedicated to a type 2B VWD patient who generously donated plasma for the study but passed away recently due to illness. We thank the Emory Children's Pediatric Research Center Flow Cytometry Core and Emory Integrated Cellular Imaging Core for technical support. This work was supported in part by National Institutes of Health grants HL082808 and HL123984 (R.L.), a Bridge Grant from the American Society of Hematology (R.L.) and a Faculty Research Grant from Lehigh University (X.F.Z.).

## Author contributions

W.D. designed and performed the experiments, analysed the data, prepared the figures and wrote the manuscript; Y.X. and M.A.D. performed single-molecule force measurements; W.C. and D.S.P. performed platelet clearance studies; A.K.S. assisted in the uniform shear assay; X.L. purified botrocetin and assisted in the botrocetin-binding assay; P.Z. and C.B.D. helped with inducible expression of GPIb–IX in CHO cells, and edited the manuscript; M.C.B., J.W., J.D.P. and F.L. provided critical reagents and discussed the results; W.B. and X.F.Z. analysed the data, prepared the figures and edited the manuscript; R.L. designed the study, analysed the data, prepared the figures and wrote the manuscript.

## Additional information

**Competing financial interests:** The authors declare no competing financial interests.

