## [Peer Review File · Nature Communications]

Reviewers' comments:

Reviewer #1 (expert in platelet biology and signaling)

Remarks to the Author:

In this manuscript Deng and coworkers use a model of botrocetin- and shear stress-induced VWF binding to glycoprotein Ibalpha to demonstrate that the recently identified mechanosensitive domain within GPIbalpha unfolds by VWF binding resulting in outside-in signalling. In addition, using mouse models, the authors demonstrate that botrocetin-and-shear-activated platelets are instantly cleared, which may be related to mechanosensitive domain unfolding as platelet carrying a chimeric GPIbalpha lacking the mechanosensitive domain is cleared much faster compared to wild-type platelets.

The authors have used clever and state-of-the-art approaches to demonstrate that botrocetin-and-shear-activated platelets transmit signals in response to the unfolding of the mechanosensitive domain. I think these experiments are convincing and have only a couple of relatively minor comments and questions:

1) On page 5 the authors show that a VWF mutant that binds platelets without the requirement for botrocetin acts identically as normal platelets with addition of botrocetin. Thus, also in absence of a non-physiological stimulator of platelet-VWF binding, signaling events can be demonstrated. I am not sure whether this experiment truly elucidates whether botrocetin and shear mimic 'a physiological or pathophysiological scenario'. I would advise to rephrase relevant sections on page 5. In addition, although I fully understand the reasons to choose for the botrocetin/shear approach under conditions in which platelets do not aggregate or agglutinate, I wonder whether it wouldn't be possible to validate key findings using immobilised VWF (to which platelets adhere without requirement for botrocetin).

2) I somewhat struggle with the interpretation of figure 2. The authors should better explain to what extent this A1 construct binds platelets spontaneously, and how botrocetin contributes to binding in this setting. Additionally, the 'MSD unfolding event in the pulling curve' needs to be better explained.

3) On page 6 the authors suggest that GPIbalpha shedding by botrocetin and shear is not mediated by ADAM17. The authors should provide more definitive proof by using ADAM17 inhibitors, and should

consider alternative known GPIIb/IIIa sheddases (plasmin, cathepsin, etc). Given additional proteases that have been shown to proteolyse GPIIb/IIIa, I wonder whether it is correct to speak about 'THE shedding cleavage site' on page 6.

The in vivo part of the work contains solid experiments, but the interpretation of these data is less straightforward than the interpretation of the in vitro work. Although the work is highly suggestive of a role of unfolding of glycoprotein Ib in platelet clearance, the authors appear to ignore alternative options.

1) Have the authors shown that the CFSE label remains inside the platelet after shear and botrocetin treatment? As this treatment results in P-selectin exposure, I assume that a general secretion reaction also occurs (which could lead to release of (part of the) CFSE label)).

2) The authors infuse partially activated, partially p-selectin positive platelets. It is well known that activated platelets are rapidly cleared from circulation. Could the clearance pattern in figure 5E not be fully independent of GPIIb/IIIa? Similarly, if the II4R chimeric mice have filopods in their resting state, they're likely to be slightly activated, which may contribute to their clearance. What about p-selectin, ECL, FURA2 values in chimeric vs wild-type mice? Does the reduced glycosylation of the chimera not suggest that the clearance mechanism of these platelets are via a different route than the Ashwell-Morell receptor?

In general, the paper is very well written. The abstract and introduction are flawless, and the results section requires minor modification as indicated earlier. I like the first part of the discussion, in which limitations of the current GPIIb/IIIa clustering model are outlined, and I think the authors have provided strong evidence for their "trigger" model. I also like the end of the discussion in which the newly identified mechanism of MSD unfolding is discussed in a broader perspective. Although I fully agree with the conclusions on the implications for this work in understanding VWF-platelet interaction, I am not fully convinced that the in vivo experiments presented are truly compatible (yet) with a critical role of MSD unfolding in platelet clearance (although I admittedly do not know how to design experiments that would convince me). I would like the authors to consider adding comments of caution on this in the discussion (would they agree with my assessment).

Reviewer #2 (platelets biology and signaling)

Remarks to the Author:

Previously, using single molecule measurements of the pulling force of the VWF A1 domain on immobilized recombinant full-length GPIb-IX complex, the same group has shown that a region within the extracellular juxtamembrane stalk of GPIb α , called the mechanosensitive domain (MSD, Ala417-Phe483), unfolds and extends upon VWF A1-dependent pulling of GPIb α . In this manuscript by Deng et al, the authors attempt to extend this work by suggesting that physiological shear in the presence of VWF and botrocetin also induce unfolding of the platelet GPIb α MSD, and that these events trigger intracellular signaling in intact platelets. It has previously been established that GPIb mediates signals, leading to integrin activation. In this manuscript, however, the evidence in support of their claims of GPIb signaling and role of MSD in signaling is unconvincing. For example, it is a stretch to claim that filopodia formation in unstimulated GPIb α cells indicates GPIb signaling. There are too many unsubstantiated claims and speculations in the manuscript. There are also other problems such as lack of proper controls and interpretation of data.

Additional comments:

One reoccurring issue is that many of the studies in this manuscript involve using flow cytometry to measure platelet activation of botrocetin/VWF agglutinated platelets. In these experiments the authors do not make it clear which gating has been used to gather the data on platelet activation. Particularly since they are comparing agglutinated platelets to single platelets in the absence of botrocetin, the SSC/FSC platelet gating used to generate the fluorescence histograms should be shown in the same figure (see figure S1 panels d,e and Figure 1b). In addition, in these and other experiments the authors do not have adequate controls for non-specific fluorescence. In addition, the data suggesting GPIb-IX MSD mutant cells and IL4R-Ib α -tg platelets induce constitutive signaling independent of VWF binding can be explained by too many other possibilities. Additionally, the authors used a lot of space describing botrocetin-mediated platelet clearance, which is not new. The role of MSD in that process is also unconvincing.

Fig1b: Show the gating for these data (+/-botrocetin) in the same figure. Background and non-specific signal is unclear for these data.

Fig S1: (b) How much botrocetin was added? (c) How was VWF binding detected?

Fig S2: Gating has not been applied fairly to these data (see panel a, p-selectin, +/- shear; and panel b, fura-2 and PAC-1) and thus give false impression of the effect of botrocetin/shear.

Fig 3: Unclear what gating is being used to analyze these data.

Fig 3 i: How do the authors know the quenching effect is specific?

Is GPIb-IX signaling presented in this manuscript important for integrin activation? Is MSD important for integrin activation?

Reviewer #3 (platelets biology and GPIb)

Remarks to the Author:

How cell receptors sense force and respond is an important question.

There are only a handful of examples of proteins well characterized in terms of the effects of force which link structure to a relevant function. The best characterized shear force sensor is the VWF-A2 domain where a protease cleavage site is exposed. In the same hemostatic system the authors here Deng et al describe the receptor for VWF (GPIb) also has a force sensing domain called the MSD and describe a variety of different techniques utilized to explore the characteristics of this domain.

The hypothesis here is that a folded MSD sequence locks the GpIb receptor into an "off" state and when this unfolds signaling, shedding and platelet clearance are activated. Overall the paper is well written and there is a variety of approaches utilized to investigate a unique phenomenon.

The shear sensor can only be located in the multimeric VWF as this protein forms a large fibre that can be affected by shear. A single protein like the gpIb receptor is not going to be affected by liquid shear. In my opinion the gpIb MSD is a VWF sensor rather than a shear sensor.

There is large difference in the mouse and human vascular systems in the sense the flow of blood in a mouse is an order of magnitude faster than a human.

When they say clustering of the receptor is not sufficient for signaling does this also mean it is not required at all? The GpIb cytodomain has a well characterised interaction with filamin (2006 107: 1925-1932) which is a dimeric protein so it seems that the receptor is likely to be a dimer.

The difference with VWFA2 is there is no crystal structure of the MSD and the complexity of the system is such that this "folded" region from the alpha subunit is likely buried in complex with the folded domains of the GPIbbeta and GPIX subunit ectodomains.

How does this research relate to the platelet filopodia form tether like extensions from the platelet .
Blood. 2011 Mar 3;117(9):2718-27..

The authors should mention the well characterized gpIb-filamin interaction Blood. 2011 Mar
3;117(9):2718-27.

Page 5. second paragraph

The authors should reference the galactose exposure in the introduction or explain this is more clearly.
Although i realize there are a lot of papers on galactose exposure this reviewer has heard of P-selectin
and calcium changes being associated with platelet activation but galactose exposure is an ambiguous
phenomenon and seems to be associated with the work of one group. Where does the galactose come
from and how it is linked with platelet activation mechanism i.e. signalling pathways within the platelet?

The idea that a shedding cleavage site becomes exposed in response to shear is very interesting.
perhaps a schematic diagram in figure 7 could be added to explain this and an illustration as to where
the cleavage site is in respective of the folded MSD. Does this link to a reversibility of platelet gpIb-VWF
tethers and filopodia extensions or is the shedding a different process?.

Page 6.

"shear did not reduce expression levels of other substrates?"

Not clear why this is an argument for ADAM17 not being the protease.

Minor comments

I don't find this a useful term as throughout the manuscript the term trigger is used in regards of
triggering cellular signalling and I get this confused with the reference to the "trigger"sequences

The terminology "Trigger" is also not clear and I see it described in figure S7. I think this figure should be brought into the main body of the paper to show the cleavage site as well as MSD and "trigger". I think as the trigger is an amino acid motif it should be called trigger sequence

The Figure S7 title is not really accurate as the MSD is not well conserved at all compared to the sequences in the beta and ix ectodomains. Only the trigger sequence is conserved features. Please indicate where the cleavage site is. The authors should state how the alignment was made and the basis of the coloring. better to use the boxshade server.

The trigger sequence is prior to the macro-glycopeptide which has many O-linked glycosylated sites. Is the MSD O-glycosylated? I notice a residue in the trigger sequence is conserved as a Ser/Thr.

Page 7

"Postulating that upon the unfolding of the MSD

insert the

Page 8.

filapodia extension in CHO cells is the same as platelets

I can understand use of the Bc abbreviation in the figures but it is not helpful to abbreviate botrocetin to Bc in the main text. Throughout the literature it is termed just botrocetin and I found it difficult to read abbreviated as Bc which reads like the name of a cell line.

Reviewer #4 (expert in platelets biology)

Remarks to the Author:

Extensive data are presented that build upon their story that VWF binding GpIb α under shear exerts a pulling force that unfolds a signal transducing juxtamembranous domain (the "MSD").

Major comments:

Please explain the rationale for using botrocetin and its mechanism of effect.

Do higher shear stresses (supraphysiological) without botrocetin recapitulate responses observed with botrocetin and low shear stresses?

Other comments:

Figure 1a - point out macroglycopeptide region

Suppl Fig 1 - how is VWF binding measured? Can VWF binding be measured at higher shear stresses without botrocetin (like in *Biorheology*. 1997 Jan-Feb;34:57-71)

There is evidence that shear-induced binding of VWF to GpIb α increases intracellular calcium via an influx pathway. If PRP is recalcified but aggregation is inhibited by another means, does ECL binding (β galactose expression) change? Are there data to determine if extracellular calcium has any effect on unfolding of the MSD?

GpIb α is also down-regulated by activation-driven internalization (for example *Blood*. 1996 Jan 15;87:618-29); how does one sort out this effect from EDTA-inhibited cleavage?

Although botrocetin-mediated platelet clearance is interesting, its *in vivo* relevance is uncertain. In contrast, shear-induced platelet aggregation is probably important for *in vivo* thrombosis. Therefore, isn't it of interest to determine if the MSD directs GpIIb-IIIa activation?

Figure 5d legend and text - please explain the JON/A reagent

Figure 6 - chimera experiments may be distracting and susceptible to misinterpretation. Why would a fixed unfolding - rather than literal traction or pull exerting force through the membrane on the cytoplasmic domain of GpIb α attached to the cytoskeleton - cause constitutive signaling? The CHO cell experiments also raise this concern and introduce the possibility that the MSD effects clustering (despite its dismissal in para 1 of the discussion) leading to signaling rather than traction leading to signaling. Some discussion of these ambiguities is needed.

Response to the Reviewers

We would like to thank each of the reviewers for their time in reviewing this manuscript. We appreciate their feedback and constructive appraisal of our manuscript. Please see our response to the reviewers' comments (in *italics*) below.

Reviewers' comments:

Reviewer #1 (expert in platelet biology and signaling)
Remarks to the Author:

In this manuscript Deng and coworkers use a model of botrocetin- and shear stress-induced VWF binding to glycoprotein Iba α to demonstrate that the recently identified mechanosensitive domain within GPIba α unfolds by VWF binding resulting in outside-in signalling. In addition, using mouse models, the authors demonstrate that botrocetin-and-shear-activated platelets are instantly cleared, which may be related to mechanosensitive domain unfolding as platelet carrying a chimeric GPIba α lacking the mechanosensitive domain is cleared much faster compared to wild-type platelets.

The authors have used clever and state-of-the-art approaches to demonstrate that botrocetin-and-shear-activated platelets transmit signals in response to the unfolding of the mechanosensitive domain. I think these experiments are convincing and have only a couple of relatively minor comments and questions:

1) On page 5 the authors show that a VWF mutant that binds platelets without the requirement for botrocetin acts identically as normal platelets with addition of botrocetin. Thus, also in absence of a non-physiological stimulator of platelet-VWF binding, signaling events can be demonstrated. I am not sure whether this experiment truly elucidates whether botrocetin and shear mimic 'a physiological or pathophysiological scenario'. I would advise to rephrase relevant sections on page 5. In addition, although I fully understand the reasons to choose for the botrocetin/shear approach under conditions in which platelets do not aggregate or agglutinate, I wonder whether it wouldn't be possible to validate key findings using immobilised VWF (to which platelets adhere without requirement for botrocetin).

Response 1: Following the reviewer's suggestion, we have revised substantially the first part of the Results section on pages 4 and 5. At the beginning (page 4), we added the following sentences to explain the rationale for using botrocetin and its mechanism of effect: "To test whether GPIb-IX can respond to physiological shear stress and induce signaling in the platelet, we first sought to establish in the lab an experimental system in which VWF binding to GPIba and shear stress within the physiological range (0-25 dyn/cm²) could be achieved. Since many conditions under which VWF is induced to bind GPIba are complicated and may contain elements beyond physiological range, botrocetin, a snake venom C-type lectin that induces binding of plasma VWF to platelets in the absence of shear through its simultaneous interactions with the A1 of VWF and the LBD of GPIba, was used in this study." In the second paragraph (page 5), we removed the phrase "to test whether the Bc/shear treatment mimics a physiological

or pathological scenario”. Instead, we stated that “Spontaneous binding of VWF to GPIIb/IIIa also occurs in many patients with type 2B von Willebrand disease (VWD)”, and that VWF.V1316M and shear induced similar signaling as botrocetin/shear.

As to this reviewer’s suggestion “to validate key findings using immobilized VWF”, we had actually considered it during the study. We decided not to do it because this experiment would not provide definitive answers for our study. But before spelling out detailed reasons below, I would like to note that a similar assay (platelet flowing over immobilized recombinant A1 domain of VWF) had been published in 2014 by Matt Auton’s group; their finding is remarkably consistent with ours. We cited their work in the original manuscript.

A key finding of our study is that the physiological, not elevated, shear stress is sufficient to induce MSD unfolding on the platelet. Although it is feasible to flow platelets over immobilized VWF in a flow chamber at a physiological shear rate, the non-uniform nature of the immobilization may not exclude the possibility that the elevated shear force may be transiently present. Moreover, in our study MSD unfolding was detected by fluorescence quenching of antibodies attached to the MSD. It would be technically challenging to detect and quantitate FRET under flow because (1) not all platelets may be attached at the same time and (2) the solid surface on which VWF is immobilized is inherently anisotropic. Finally, VWF is a multi-domain protein. Other domains than A1 may potentially participate in adhesion and signaling, thus complicating data interpretation. On the last note, Auton’s group has monitored the flow of platelets over the immobilized A1 domain instead of full-length VWF (Biophys. J. 107: 1185, 2014; cited at ref #59 in the revised manuscript). They observed that platelets “stop-and-go” on the A1-coated surface under flow. For A1 domains with various type 2B mutations, the pause time, the average time a platelet stops on a mutant A1 surface before rolling again, is correlated with the severity of thrombocytopenia in VWD patients bearing the same mutation. In my opinion, the pause time in Auton’s study should correlate with the unbinding force of the A1/GPIIb-IIIa interaction in our study. The longer the pause time, the stronger the unbinding force, thus more likely the induction of MSD unfolding. Their finding is consistent with our results in this manuscript.

2) I somewhat struggle with the interpretation of figure 2. The authors should better explain to what extent this A1 construct binds platelets spontaneously, and how botrocetin contributes to binding in this setting. Additionally, the 'MSD unfolding event in the pulling curve' needs to be better explained.

Response 2: Following the reviewer’s suggestion, the description of Figure 2 results (p.6, 2nd paragraph) has been substantially revised. That “recombinant A1 could bind the LBD and platelets spontaneously” was added, and botrocetin’s enhancement of the A1/LBD interaction described. Also, a few sentences were added to describe the single-molecule force experiment in more detail and to explain why botrocetin could help to induce MSD unfolding more frequently by increasing the unbinding force and the bond lifetime.

3) On page 6 the authors suggest that GPIIb/IIIa shedding by botrocetin and shear is not mediated by ADAM17. The authors should provide more definitive proof by using ADAM17 inhibitors, and should consider alternative known GPIIb/IIIa sheddases (plasmin, cathepsin,

etc). Given additional proteases that have been shown to proteolyse GPIbalpha, I wonder whether it is correct to speak about 'THE shedding cleavage site' on page 6.

Response 3: As suggested, we have added a new piece of data into Figure 3b – like EDTA, addition of GM6001, a broad-spectrum metalloprotease inhibitor, also completely inhibited the down-regulation of GPIba level induced by the botrocetin/shear treatment. This provides additional supporting evidence that the down-regulation was due to metalloprotease-mediated shedding of GPIba.

In addition, we have revised the manuscript (p.7, 2nd paragraph) to clarify something about ADAM17 and metalloproteases in general. ADAM17 has a basal or constitutive activity; that's why GPIba is continuously shed from the resting platelet. But the ADAM17 activity can be upregulated upon stimulation. The same can be said about other metalloproteases. In our system, GPIba is still being continuously shed from the platelet. The reason we did not talk about this basal shedding activity is because it happens on a time scale much slower than our botrocetin/shear study. It was the increase in GPIba shedding that was investigated in Figure 3. Thus, our data do not suggest that botrocetin/shear-induced GPIba shedding is not mediated by ADAM17. Rather our data suggest that botrocetin/shear-induced GPIba shedding is not caused by an increase in the ADAM17 activity (because we did not observe a similar increase in shedding of other membrane proteins). It should be noted here that GPV is also continuously shed from the platelet. Nonetheless, this reviewer is correct in that our results could not rule out the possibility that other metalloproteases than ADAM17 may participate in shedding of GPIba, since these proteases are notoriously known for their broad substrate specificity. With this in mind, we have revised the last sentence of the paragraph to “....., suggesting that botrocetin/shear induced GPIba shedding via a mechanism that does not involve the activation of ADAM17 or other metalloproteases.” And “the shedding cleavage site” in the manuscript is changed to “the ADAM17 cleavage site”.

The in vivo part of the work contains solid experiments, but the interpretation of these data is less straightforward than the interpretation of the in vitro work. Although the work is highly suggestive of a role of unfolding of glycoprotein Ib in platelet clearance, the authors appear to ignore alternative options.

1) Have the authors shown that the CFSE label remains inside the platelet after shear and botrocetin treatment? As this treatment results in P-selectin exposure, I assume that a general secretion reaction also occurs (which could lead to release of (part of the) CFSE label)).

Response 4: CFSE is a membrane permeable fluorescent dye that covalently conjugates with the primary amine group of proteins on the cell surface and in the cytoplasm. Thus, even with the secretion of granules, the majority of CFSE label should have remained with the platelet. Indeed this was what we observed when we measured and compared the CFSE levels in the platelet before and after botrocetin/shear (added as Figure S9a in the revised manuscript). Also, it should be noted that the extent of botrocetin/shear-induced P-selectin expression is only ~5-10% of that induced by thrombin activation. Thus we don't think there is a general secretion reaction following the botrocetin/shear treatment.

2) The authors infuse partially activated, partially p-selectin positive platelets. It is well known that activated platelets are rapidly cleared from circulation. Could the clearance pattern in figure 5E not be fully independent of GPIb α ? Similarly, if the IL4R chimeric mice have filopods in their resting state, they're likely to be slightly activated, which may contribute to their clearance. What about p-selectin, ECL, FURA2 values in chimeric vs wild-type mice? Does the reduced glycosylation of the chimera not suggest that the clearance mechanism of these platelets are via a different route than the Ashwell-Morell receptor?

Response 5: The rapid clearance shown in Figure 5E was a consequence of botrocetin/shear treatment. Its effect on platelets was initiated with binding of botrocetin/VWF to GPIb-IX. Therefore I believe that the clearance pattern should be dependent on botrocetin/VWF. But there is a possibility that botrocetin/VWF itself also induces uptake by macrophages as described in previous studies. This may help to account for the difference in the extent of clearance in vivo between botrocetin/shear-treated platelets and IL4R-Ib α Tg platelets. In the revamped Discussion, we have acknowledged this possibility (p.15).

As suggested by this reviewer, we have compared P-selectin expression, ECL binding and Fura2 fluorescence in IL4R-Ib α Tg platelets to those in the WT. The new data, included as Figure 6d and Supplement Figure 10a, provide additional supporting evidence that the IL4R-Ib α Tg platelets are actually slightly activated from GPIb-IX signaling. We have revised the manuscript to incorporate these data (p.11, 2nd paragraph).

In our trigger model, MSD unfolding induces GPIb-IX signaling, leading to the presentation of “clear-me” signs on the platelet surface and subsequent clearance. It is possible that P-selectin expression, exposure of β -galactose, calcium influx, or filopodia formation may be involved in mediating platelet clearance, but the intent of this manuscript is not to ascertain their roles in platelet clearance or the “clear-me” signs on the platelet. In this manuscript we utilized them only as the downstream indicators of GPIb-IX signaling, but we did acknowledge the likely role of β -galactose exposure in mediating platelet clearance in the Discussion.

I assume what this reviewer meant by “the reduced glycosylation of the chimera” is that, since the GPIb α extracellular domain is so heavily glycosylated, without it the IL4R-Ib α Tg platelet should be less glycosylated than the wild type. It makes sense, although there has not been any supporting evidence reported. We have compared the extent of ECL binding (i.e. exposure of β -galactose) of WT and IL4R-Ib α Tg platelets and found no statistical difference between them. Since the IL4R-Ib α Tg platelet may be less glycosylated than the WT, it is possible that the ECL binding level in the IL4R-Ib α Tg platelet reflects the reduced level of total glycans and simultaneous increased exposure of β -galactose. Unfortunately we do not have appropriate tools to dissect it further to reach a conclusion on the ECL binding result.

While it is not determined whether botrocetin/VWF/shear-stimulated platelets are cleared through recognition by the Ashwell-Morell receptor or a different route, we had noted in the original manuscript that anti-LBD antibody-induced platelet clearance is mediated, at least partly, through the Ashwell-Morell receptor (ref #56 in the revised manuscript), and that botrocetin/shear induced similar signaling events (e.g. P-selectin expression) in the platelet as anti-LBD antibodies. Although the involvement of Ashwell-Morell receptor in platelet clearance

is demonstrated using knockout animals, details of the underlying mechanism remain to be clarified. For instance, it was suggested that the Ashwell-Morell receptor recognizes the exposed β -galactose on the LBD of GPIba, because proteolytic removal of the LBD from the platelet surface by O-sialoglycoprotein endopeptidase abolished the clearance of treated platelets by Ashwell-Morell receptor (Figure 5a in Nat. Med. 15(11): 1273–1280, 2009). It was reasoned that since removal of the LBD includes the 2 N-glycans on the LBD, it should be the galactoses on those 2 N-glycans that are recognized by the Ashwell-Morell receptor. However, while the LBD of human GPIba contains 2 N-glycans, that of murine GPIba contains no N-glycosylation sequence motifs (i.e. NxS/T) and thus should contain no N-glycans. And the above experiment was performed using murine platelets, not human ones! In other words, removal of the LBD that contained no glycans abolished the clearance by Ashwell-Morell receptor, which clearly is not consistent with the idea that the Ashwell-Morell receptor recognizes the exposed β -galactose on the LBD. Thus, while I am convinced that the glycan change and the Ashwell-Morell receptor are critical for platelet clearance, I am not convinced that the detailed mechanism of Ashwell-Morell receptor-mediated platelet clearance is fully elucidated yet. It is interesting to ponder how our trigger model fits with the model of Ashwell-Morell receptor, but this topic is beyond the scope of this manuscript.

In general, the paper is very well written. The abstract and introduction are flawless, and the results section requires minor modification as indicated earlier. I like the first part of the discussion, in which limitations of the current GPIbalpha clustering model are outlined, and I think the authors have provided strong evidence for their "trigger" model. I also like the end of the discussion in which the newly identified mechanism of MSD unfolding is discussed in a broader perspective. Although I fully agree with the conclusions on the implications for this work in understanding VWF-platelet interaction, I am not fully convinced that the in vivo experiments presented are truly compatible (yet) with a critical role of MSD unfolding in platelet clearance (although I admittedly do not know how to design experiments that would convince me). I would like the authors to consider adding comments of caution on this in the discussion (would they agree with my assessment).

Response 6: In response to this reviewer's suggestion and also to related questions from other reviewers, we have substantially revised the Discussion by adding two paragraphs. Both paragraphs cover a large number of previous observations that connect GPIb-IX and GPIba shedding to platelet clearance and thrombocytopenia. In one paragraph that follows the description of our trigger model (p.12, 3rd paragraph), we discussed in detail how our findings, particularly those of type 2B VWD and exposure of β -galactose, fit with recent literature on desialylation, Ashwell-Morell receptor and anti-LBD antibody-mediated platelet clearance. As a note of caution we also discussed the "unknown links" between MSD unfolding-induced GPIb-IX signaling and presentation of "clear-me" signs. In the other paragraph (p.14, 2nd paragraph), we discussed in detail the results of CHO-Iba Δ /Ib β /IX cells and IL4R-IbaTg platelets. While similarities in GPIb-IX signaling between these cells and botrocetin/shear-treated platelets are noted, a note of caution is added to discuss the small difference in the extent of in vivo clearance between them. Overall, I hope these additions in the Discussion will help to convey that the connection between GPIb-IX and platelet clearance has been very well documented, and that the idea of MSD unfolding to induce GPIb-IX signaling and platelet clearance is compatible with the literature.

Reviewer #2 (platelets biology and signaling)
Remarks to the Author:

Previously, using single molecule measurements of the pulling force of the VWF A1 domain on immobilized recombinant full-length GPIb-IX complex, the same group has shown that a region within the extracellular juxtamembrane stalk of GPIb α , called the mechanosensitive domain (MSD, Ala417-Phe483), unfolds and extends upon VWF A1-dependent pulling of GPIb α . In this manuscript by Deng et al, the authors attempt to extend this work by suggesting that physiological shear in the presence of VWF and botrocetin also induce unfolding of the platelet GPIb α MSD, and that these events trigger intracellular signaling in intact platelets. It has previously been established that GPIb mediates signals, leading to integrin activation. In this manuscript, however, the evidence in support of their claims of GPIb signaling and role of MSD in signaling is unconvincing. For example, it is a stretch to claim that filopodia formation in unstimulated GPIb Δ cells indicates GPIb signaling.

Response 7: Filopodia formation in platelets and transfected CHO cells expressing GPIb-IX has been used to indicate GPIb-IX signaling for more than 15 years and reproduced in several reputable independent labs around the world (e.g. Shaun Jackson lab, Francois Lanza lab, Steve Watson lab). Key supporting evidence includes that mutations in the GPIb α or GPIb β cytoplasmic domain affect the filopodia formation, and that anti-GPIb β antibody RAM.1 blocks the filopodia formation but does not affect the attachment of platelets or CHO cells to VWF.

We have performed additional experiments and found that RAM.1, but not rat IgG, inhibited all filopodia formation in iCHO-Ib α Δ /Ib β /IX cells. This result is now included as Figure 4gh in the revised manuscript (p.9, 1st paragraph) to provide additional evidence supporting our conclusion that the unfolded MSD emits a signal through GPIb-IX into the cell.

There are too many unsubstantiated claims and speculations in the manuscript. There are also other problems such as lack of proper controls and interpretation of data.

Response 8: We have gone through the manuscript and added back controls in both supplementary and main figures:

- 1, Non-specific fluorescence controls in Figure 1b;*
- 2, Gating information in Figure S1e;*
- 3, Controls demonstrating the FRET specificity in Figure S5a,b;*
- 4, Non-specific fluorescence controls in Figure S1c;*
- 5, Raw flow cytometry data of botrocetin/shear-treated murine platelets as Figure S8.*

Additional comments:

One reoccurring issue is that many of the studies in this manuscript involve using flow cytometry to measure platelet activation of botrocetin/VWF agglutinated platelets. In these experiments the authors do not make it clear which gating has been used to gather the data on platelet activation. Particularly since they are comparing agglutinated platelets to single platelets in the absence of botrocetin, the SSC/FSC platelet gating used to generate the fluorescence histograms should be shown in the same figure (see figure S1 panels d,e and Figure 1b). In addition, in these and other experiments the authors do not have adequate

controls for non-specific fluorescence.

Response 9: To address this reviewer's concern, we added a paragraph in the Methods section (p.29, 2nd paragraph) describing the flow cytometry procedure in detail including the instrument, the gate setting, and other related information. We also added gating information to Figure S1e. The same gating was applied to all the data shown in the figures. Non-specific fluorescence controls in Figure 1b have been added back. There was a mistake in the histogram of Fura-2 (+Bc) in Figure 1b, which has been corrected in the revised manuscript. Overall, our conclusion remains the same.

In addition, the data suggesting GPIb-IX MSD mutant cells and IL4R-Ib α -tg platelets induce constitutive signaling independent of VWF binding can be explained by too many other possibilities.

Response 10: It is difficult to address this broad critique, but I hope the aforementioned RAM.1 inhibition of filopodia formation in iCHO-Ib α Δ /Ib β /IX cells would help to allay this reviewer's suspicions.

Additionally, the authors used a lot of space describing botrocetin-mediated platelet clearance, which is not new. The role of MSD in that process is also unconvincing.

Response 11: Yes botrocetin-mediated platelet clearance was first reported in 1980's by the Brinkhous group (cited in the original manuscript), but the underlying molecular mechanism has never been elucidated. Our study is the first to provide evidence for a mechanism at the molecular level. Since Brinkhous group reported the phenomenon in other animals but not in mice, we were obligated to start from scratch and report details of botrocetin-mediated acute platelet clearance in mice.

Fig1b: Show the gating for these data (+/-botrocetin) in the same figure. Background and non-specific signal is unclear for these data.

Response 12: Please see Response 9 above.

Fig S1: (b) How much botrocetin was added? (c) How was VWF binding detected?

Response 13: We have added the information to the Methods section and legend of Figure S1: The botrocetin concentration used was 2 μ g/mL. VWF binding was detected using a rabbit anti-VWF polyclonal antibody followed by a fluorescent donkey anti-rabbit secondary antibody.

Fig S2: Gating has not been applied fairly to these data (see panel a, p-selectin, +/- shear; and panel b, fura-2 and PAC-1) and thus give false impression of the effect of botrocetin/shear.

Response 14: The MFI values shown in Figure 1c are of the entire population without any additional gating applied. For Figure 1b, we have followed the reviewer's suggestion and applied the same gating for the data. The results shown in the updated figures still support our original conclusion that both botrocetin and physiological shear stress are required to induce

platelet signaling.

Fig 3: Unclear what gating is being used to analyze these data.

Response 15: The same SSC/FSC gating shown in Figure S1e is applied to the data shown in Figure 3. We have clarified this point in the revised manuscript.

Fig 3 i: How do the authors know the quenching effect is specific?

Response 16: We have added two figures as Figure S5 to demonstrate the specificity of F-5G6 quenching by Q-RAM.1. First, 5G6 binding to the platelet is the same as that in the presence of unlabeled RAM.1. Second, fluorescence of fluorescently labeled WM23, which binds an epitope in the nearby macroglycopeptide region, is not quenched by Q-RAM.1.

Is GPIb-IX signaling presented in this manuscript important for integrin activation? Is MSD important for integrin activation?

Response 17: It is potentially important. Whether the MSD is important for integrin activation is beyond the scope of this manuscript. We have revised the Discussion (p.16, 2nd paragraph) to clarify what we found in this study and also note that our results may facilitate future studies on interplays between platelets, plasma and vessels under diverse shear flow conditions.

Reviewer #3 (platelets biology and GPIb)

Remarks to the Author:

How cell receptors sense force and respond is an important question.

There are only a handful of examples of proteins well characterized in terms of the effects of force which link structure to a relevant function. The best characterized shear force sensor is the VWF-A2 domain where a protease cleavage site is exposed. In the same hemostatic system the authors here Deng et al describe the receptor for VWF (GPIb) also has a force sensing domain called the MSD and describe a variety of different techniques utilized to explore the characteristics of this domain.

The hypothesis here is that a folded MSD sequence locks the GPIb receptor into an "off" state and when this unfolds signaling, shedding and platelet clearance are activated. Overall the paper is well written and there is a variety of approaches utilized to investigate a unique phenomenon.

The shear sensor can only be located in the multimeric VWF as this protein forms a large fibre that can be affected by shear. A single protein like the GPIb receptor is not going to be affected by liquid shear. In my opinion the GPIb MSD is a VWF sensor rather than a shear sensor.

Response 18: Whether the MSD is a VWF sensor, or a shear sensor, or both may be up for debate, but we showed in this manuscript that both VWF binding and physiological shear are required for inducing MSD unfolding on the platelet surface. By mentioning "a single protein like the GPIb receptor is not going to be affected by liquid shear", this reviewer raised a very

good question on how physiological shear and binding of soluble ligands combined to exert a pulling force on GPIb-IX. We have our model and we are testing it, but it would be beyond the scope of this manuscript to include the answer here.

There is large difference in the mouse and human vascular systems in the sense the flow of blood in a mouse is an order of magnitude faster than a human.

Response 19: There may be a difference in flow between mouse and human, but the difference does not seem to critically affect our results and impact the conclusion of our study. We showed in the manuscript that both human and murine platelets can be stimulated by botrocetin/shear, in the same shear range. In addition, thrombocytopenia is observed in both VWD type 2B patients and mice expressing VWF with the same type 2B mutation.

When they say clustering of the receptor is not sufficient for signaling does this also mean it is not required at all? The GpIb cytodomain has a well characterised interaction with filamin (2006 107: 1925-1932) which is a dimeric protein so it seems that the receptor is likely to be a dimer.

Response 20: We did not mean “not required at all”, because there is no clear evidence addressing this issue. But in my opinion it is clear that the current clustering model (J. Thromb. Haemost. 11(supple 1): 330-9, 2013) is not adequate to explain the shear requirement. However, this is a complicated issue; even GPIb dimer may mean different things to different investigators. For instance, anti-LBD antibody-crosslinked GPIb dimer on the platelet surface may not be the same as filamin-crosslinked GPIb dimer. Considering that we do not have evidence in this manuscript addressing the clustering model, and to avoid potential confusion by the readers, we have removed the sentences from the Discussion that were related to the clustering model (p.12). The relationship between the trigger model proposed here and the clustering model in the literature is a topic for another manuscript.

The difference with VWFA2 is there is no crystal structure of the MSD and the complexity of the system is such that this "folded" region from the alpha subunit is likely buried in complex with the folded domains of the GPIbeta and GPIX subunit ectodomains.

Response 21: We agree with the reviewer’s assessment that it forms direct contacts, probably extensive contacts, with GPIb β and/or GPIX extracellular domains. But it should be noted that a portion of it, at least the ADAM17 cleavage site, is on the surface of folded MSD and accessible to antibody binding (Sci. Rep. 6: 24789, 2016).

How does this research relate to the platelet filopodia form tether like extensions from the platelet . Blood. 2011 Mar 3;117(9):2718-27.

The authors should mention the well characterized gpib-filamin interaction Blood. 2011 Mar 3;117(9):2718-27.

Response 22: We have read the paper mentioned by this reviewer and cited it in our revised manuscript. The paper demonstrated the importance of the GPIba/filamin interaction under high shear stress (e.g. shear rate of 10,000 s⁻¹). Disrupting the GPIba/filamin interaction appears to

affect neither the ligand binding activity of GPIba nor adhesion/spreading of platelets over immobilized VWF in the presence of ristocetin. While these results are not inconsistent with the conclusion of our study, I am not sure how they relate to our results, which addressed only what happens within the physiological shear stress. In our study the filopodia formation is utilized only as an indicator of GPIb-IX signaling; its physiological function was not studied and should not affect the conclusion of our study. Also, it is possible that the tether-like extension in the 2011 Blood paper is structurally distinct from the filopodia we observed – the former appears to have GPIb-IX at its tip, where it binds to the immobilized ligand at the surface, whereas the filopodia we observed does not have ligand-binding at its tip. It is of interest to note that, other than the mutation to disrupt the GPIba/filamin interaction, the 2011 Blood paper did not address the dynamics of GPIb-IX under high shear flow conditions. It is my hope that our first description of the dynamic change of MSD in response to ligand binding and physiological shear will help future studies of platelets under diverse shear conditions.

Page 5. second paragraph

The authors should reference the galactose exposure in the introduction or explain this is more clearly. Although I realize there are a lot of papers on galactose exposure this reviewer has heard of P-selectin and calcium changes being associated with platelet activation but galactose exposure is an ambiguous phenomenon and seems to be associated with the work of one group. Where does the galactose come from and how it is linked with platelet activation mechanism i.e. signalling pathways within the platelet?

Response 23: Having carefully considered the suggestion, we decide to leave the topic of galactose exposure out of the Introduction section. As we elaborated in Response 5, the mechanism of glycan changes and Ashwell-Morell receptor in mediating platelet clearance is still not fully settled. Instead we described and cited the exposure of galactose in relation to platelet clearance in more detail in a separate paragraph in the Discussion (p.13, 2nd paragraph).

This reviewer raised a very good question on the origin of the exposed galactose and how it is linked with platelet signaling and clearance. That is the open question for the field; I don't have a clear answer for it. Although our results are consistent with the idea that exposed galactose, or a change of glycans on the platelet surface, leads to platelet clearance, how the exposed galactose leads to platelet clearance is not what we aimed to address in our manuscript nor central to the conclusion of our manuscript. To avoid potential confusion, we decide to only acknowledge in the Discussion the connection of our findings with previous reports of galactose exposure in the context of platelet clearance, and not to speculate extensively on the topic of galactose exposure.

The idea that a shedding cleavage site becomes exposed in response to shear is very interesting. perhaps a schematic diagram in figure 7 could be added to explain this and an illustration as to where the cleavage site is in respective of the folded MSD.

Response 24: We thank this reviewer for the suggestion. Additional diagram illustrating shear-induced unfolding of MSD, in which the ADAM17 cleavage site in the folded and unfolded MSD is marked, is added as Figure 3e. Adding it to Figure 7 would leave the diagram too small to see. In addition, we have marked the cleavage site in the MSD sequence now included as Figure 4a.

As a side note, I would like to clarify a small point here. What we found is that the ADAM17 cleavage site becomes more exposed in response to ligand/shear. GPIba is continuously shed from the platelet surface. The ADAM17 cleavage site in the MSD of GPIba is exposed, at least part of the time, on the platelet surface.

Does this link to a reversibility of platelet gpIb-VWF tethers and filopodia extensions or is the shedding a different process?

Response 25: I am not sure how this relates to the tether described in the 2011 Blood paper. As to the filopodia observed in our study, I consider it simply as a downstream consequence of GPIb-IX signaling. Also I like to think shedding of GPIba, which by the way is a physiological process, as an alternative pathway (i.e. alternative to ligand/shear) to induce unfolding of the Trigger sequence. In other words, iCHO-Iba Δ /Ib β /IX cells and IL4R-IbaTg platelets may mimic platelets with shed GPIba, because after GPIba shedding MSD residues that remain on the platelet surface include the Trigger sequence (please see revised Figure 4a). As only half (or less than half) of a structural domain, these residues should be unfolded. Thus, GPIba shedding should induce GPIb-IX signaling. Consistently, there are plenty of papers in the literature reporting the correlation between GPIba shedding, the extent of platelet storage lesion, and fast clearance of platelets with the storage lesion. To clarify this point, we have added a separate paragraph in the Discussion (p.14-p.15).

Page 6.

"shear did not reduce expression levels of other substrates?"

Not clear why this is an argument for ADAM17 not being the protease.

Response 26: This question is very similar to one of Reviewer 1's questions. Please see Response 3 above.

Minor comments

I don't find this a useful term as throughout the manuscript the term trigger is used in regards of triggering cellular signalling and I get this confused with the reference to the "trigger" sequences

Response 27: Thanks for the feedback. We have replaced "triggering" with "inducing" throughout in the revised manuscript.

The terminology "Trigger" is also not clear and I see it described in figure S7. I think this figure should be brought into the main body of the paper to show the cleavage site as well as MSD and "trigger".

Response 28: Following this suggestion, sequence alignment of the MSD is now included as Figure 4a. The Trigger sequence and the ADAM17 cleavage site are labeled in the figure.

I think as the trigger is an amino acid motif it should be called trigger sequence.

Response 29: Thanks for the suggestion. We have replaced “Trigger” with “Trigger sequence” in the revised manuscript.

The Figure S7 title is not really accurate as the MSD is not well conserved at all compared to the sequences in the beta and ix ectodomains. Only the trigger sequence is conserved features. Please indicate where the cleavage site is. The authors should state how the alignment was made and the basis of the coloring. better to use the boxshade server.

Response 30: Figure S7 is the revised Figure 4a. Its title has been revised as suggested. The sequence alignment was performed manually to avoid gaps inserted by the alignment software. The alignment is anchored at the conserved vicinal Cys residues of GPIb α . The result was colored using the BOXSHADE server and the shedding cleavage site was marked.

The trigger sequence is prior to the macro-glycopeptide which has many O-linked glycosylated sites. Is the MSD O-glycosylated? I notice a residue in the trigger sequence is conserved as a Ser/Thr.

Response 31: Yes there are quite a few Ser and Thr residues in the MSD. I believe part of MSD contains O-glycans, but we don't have any evidence.

Page 7
"Postulating that upon the unfolding of the MSD
insert the

Response 32: We have revised the manuscript following this suggestion.

Page 8.
filapodia extension in CHO cells is the same as platelets

Response 33: We don't understand what this comment is about.

I can understand use of the Bc abbreviation in the figures but it is not helpful to abbreviate botrocetin to Bc in the main text. Throughout the literature it is termed just botrocetin and I found it difficult to read abbreviated as Bc which reads like the name of a cell line.

Response 34: Thanks for the comment. We have gone through the manuscript and replaced all “Bc” in the text with “botrocetin”. “Bc” is only present in the figures as suggested.

Reviewer #4 (expert in platelets biology)

Remarks to the Author:

Extensive data are presented that build upon their story that VWF binding GpIb α under shear exerts a pulling force that unfolds a signal transducing juxtamembranous domain (the "MSD").

Major comments:

Please explain the rationale for using botrocetin and its mechanism of effect.

Response 35: This suggestion is very similar to one of Reviewer 1's question. Please see our Response 1 above.

Do higher shear stresses (supraphysiological) without botrocetin recapitulate responses observed with botrocetin and low shear stresses?

Response 36: Probably not. In higher shear stresses, additional processes may happen, and additional factors (e.g. the GPIb/filamin interaction) may become critical. Again, I hope that our first description of the dynamic change of MSD in response to ligand binding and physiological shear will be helpful for future studies of platelets under other shear conditions.

Other comments:

Figure 1a - point out macroglycopeptide region

Response 37: The macroglycopeptide region is marked in the revised Figure 1a as suggested.

Suppl Fig 1 - how is VWF binding measured? Can VWF binding be measured at higher shear stresses without botrocetin (like in Biorheology. 1997 Jan-Feb;34:57-71)

Response 38: Binding of VWF was measured using two methods: (1) by ELISA (Figure S1b): VWF was added to the immobilized GPIb-IX complex and the binding was detected using HRP-conjugated rabbit anti-VWF antibody, and (2) by flow cytometry (Figure S1c): plasma VWF binding to platelet was probed by incubating the cells with rabbit anti-VWF antibody and FITC-conjugated donkey anti-rabbit secondary antibody. The information was added to the figure legend of Figure S1. VWF binding may be measured at higher shear stress without botrocetin, but this study is not designed to investigate processes at higher shear stresses.

There is evidence that shear-induced binding of VWF to GpIb α increases intracellular calcium via an influx pathway. If PRP is recalcified but aggregation is inhibited by another means, does ECL binding (β galactose expression) change?

Response 39: Yes! We have performed the experiment as suggested: RGDS peptide was added to the PRP before recalcification to 1 mM calcium, the sample was then treated with botrocetin/shear as described. The ECL binding of the platelets was similar to that of platelets without the RGDS/calcium pretreatment. This result is now included as Supplementary Figure 3 and described in the revised manuscript (p.5, end of 1st paragraph).

Are there data to determine if extracellular calcium has any effect on unfolding of the MSD?

Response 40: No, we have not tested it. The A1/GPIb-IX interaction does not require calcium, and the MSD does not appear to contain any known calcium-binding motifs.

GpIb α is also down-regulated by activation-driven internalization (for example Blood. 1996 Jan 15;87:618-29); how does one sort out this effect from EDTA-inhibited cleavage?

Response 41: I believe the activation-driven internalization of GPIba is a different process from the metalloprotease-mediated shedding of GPIba described in Figure 3. In the 1996 Blood paper, 1 U/ml of thrombin was used to activate platelets. At this dose, thrombin activation is primarily mediated by PAR receptors and likely propagates through many intracellular signaling molecules. In comparison our case appears to be much simpler. It does not appear to involve any significant internalization of GPIba, since EDTA or GM6001 could fully inhibit the down-regulation of GPIba.

Although botrocetin-mediated platelet clearance is interesting, its in vivo relevance is uncertain. In contrast, shear-induced platelet aggregation is probably important for in vivo thrombosis. Therefore, isn't it of interest to determine if the MSD directs GpIIb-IIIa activation?

Response 42: It is, but it is of future interest and thus beyond the scope of this manuscript. This manuscript focuses on the dynamics of MSD in GPIb-IX signaling in response to physiological shear and its consequence in platelet clearance. While GPIb-IX clearly is required for primary hemostasis and GPIIb-IIIa activation in the process, our results, consistent with many other previous studies, suggest that MSD unfolding-mediated GPIb-IX signaling may be only a part of the machinery that leads to integrin activation. It is beyond the scope of this manuscript to determine whether and how MSD participates in integrin activation.

Figure 5d legend and text - please explain the JON/A reagent

Response 43: We have added the following sentence to the legend of Figure 5: "The PE-conjugated JON/A antibody selectively binds to the activated murine integrin α IIb β 3".

Figure 6 - chimera experiments may be distracting and susceptible to misinterpretation. Why would a fixed unfolding - rather than literal traction or pull exerting force through the membrane on the cytoplasmic domain of GpIba attached to the cytoskeleton - cause constitutive signaling? The CHO cell experiments also raise this concern and introduce the possibility that the MSD effects clustering (despite its dismissal in para 1 of the discussion) leading to signaling rather than traction leading to signaling. Some discussion of these ambiguities is needed.

Response 44: Thanks for the feedback here. Following the suggestion, we have added a separate paragraph in the Discussion (p.14-p.15) to explain why chimera experiments are critical to the conclusion and the implication of our study. The questions of this reviewer concentrated on the role of GPIb-IX signaling in activation of GPIIb-IIIa and under high shear stress. But this is not what we addressed in this manuscript. We focused on a different scenario. The literature linking GPIb-IX and GPIba shedding to platelet clearance is too extensive to ignore. In the context of the role of GPIb-IX in platelet clearance, iCHO-Iba Δ /Ib β /IX cells and IL4R-IbaTg platelets in a way may be considered as a model of GPIba shedding as described in Response 25. Also, under the conditions in our study were there no pulling forces exerted on GPIb-IX in iCHO-Iba Δ /Ib β /IX cells and IL4R-IbaTg platelets. For instance: at the mid-height of the cell where there is no VWF present (Fig. 4e-h), and washed IL4R-IbaTg platelets (Fig. 6b-d). Similarly, it is unlikely that in the absence of VWF binding GPIba Δ , but not wild-type GPIba, will cluster on its

own. Overall, our results are consistent with previous observations of GPIIb/IIIa shedding – GPIIb/IIIa shedding can happen in the absence of ligand binding and shear, but it can achieve the same effect to unfold the Trigger sequence as the botrocetin/shear treatment. As described in Response 25, we have clarified these points in the revised Discussion (p.14-p.15).

Reviewers' comments:

Reviewer #1 (Remarks to the Author):

I thank the authors for their clear responses to my questions and applaud the major changes made in the manuscript. I congratulate the authors with very interesting data. I have no additional comments.

Reviewer #2 (Remarks to the Author):

In rebuttal letter, the authors argue that "Filopodia formation in platelets and transfected CHO cells expressing GPIb-IX has been used to indicate GPIb-IX signaling for more than 15 years and reproduced in several reputable independent labs".

However, receptor signaling should be indicated by the biochemical/biophysical changes induced by the ligand binding to the receptor, resulting in physiological/pathological relevant cellular responses. Hence to conclude that a receptor transduces signal, one should (1) show that the event is induced by a ligand binding to the receptor, (2) the event is not caused by structural changes to cellular model system (such as that caused by mutations etc), and (3) the event is relevant to certain physiological cell responses. In the previous version, the authors did not provide evidence the membrane morphology changes (filopodia) they saw were caused by the receptor function of GPIb-IX, did not exclude the possibility of structural changes to the membranes, and did not provide a physiological signaling outcome of GPIb-IX signaling, such as integrin activation. Please note that GPIb-IX is also a structural membrane protein, which is tightly associated the cytoskeletal proteins critically important for cell membrane integrity and morphology. It is more than a possibility that a mutation in the GPIb directly affects morphology of the membranes without requiring receptor (or signaling) function of GPIb-IX (mutations or deletion in GPIb-IX in Bernard-Soulier syndrome causes dramatic platelet membrane structure and morphology changes, but one cannot conclude that it is caused by receptor signaling of GPIb-IX. Therefore, the authors' argument is not convincing.

With that being said, the authors in this revision have provided a new figure showing a monoclonal antibody against GPIIb/IIIa (alphaIIb beta3) RAM1, as previously reported, diminished filopodia in GPIb-IX-expressing cells, which is an improvement. However, GPIIb/IIIa is not the receptor subunit of GPIb-IX. A vWF

binding blocking anti-GPIb antibody will be much more convincing. Also, the possibility of RAM1 affected GPIb-IX cytoskeletal association or membrane structure has not been excluded here or in the original RAM1 paper in ATVB (which was not a convincing paper by any standard). Thus the authors should use the receptor blocking anti-GPIb antibody and show that filopodia occurs only on vWF but not other matrix proteins such as fibronectin or fibrinogen.

Because the goal of this manuscript is to study the receptor signaling mechanisms and the role of MSD in transducing such a signal, it is important to exclude constitutive changes caused by mutations or antibodies that affect membrane morphology. Unfortunately, the only positive data on signaling this manuscript present are the constitutive morphology changes.

In previous submission, the reviewer commented that "the data suggesting GPIb-IX MSD mutant cells and IL4R-Ib α -tg platelets induce constitutive signaling independent of VWF binding can be explained by too many other possibilities." In response, the authors state

that "It is difficult to address this broad critique, but I hope the aforementioned RAM.1

inhibition of filopodia formation in iCHO-Ib α Δ /Ib β /IX cells would help to allay this reviewer's suspicions."

The "too many possibilities" comment is based on the reasons explained above. Because the authors did not provide data supporting a receptor-based signaling, it is entirely possible that all kinds of structural changes induced by a mutation can be responsible for the morphological changes. For the same reason above, the conclusion of constitutive IL4-GPIb fusion protein signaling is not convincing. In fact, IL4-GPIb clearly abolished GPIb-associated functions in previous studies by investigators who made the mutant mice.

In response 17, the authors stated "Whether the MSD is important for integrin activation is beyond the scope of this manuscript." If the data supporting signaling is convincing, this reply may be ok, but when the morphological changes are not convincing to the reviewer, showing whether MSD is important for integrin activation becomes an easy way to convince the reviewer that it is indeed mediating signaling. In fact, the reviewer was suggesting how to improve the manuscript.

Reviewer #3 (Remarks to the Author):

Overall i am satisfied with the revisions and this paper does give a new way of viewing the function of a key platelet receptor and using a diverse array of techniques links the receptor unfolding mechanisms which has ramifications for understanding the widespread phenomemon of platelet clearance and also important for developing new ways of platelet storage.

minor correction

Page 15 . line 324 - insert "the"

These results also suggest that the extension of "the" Trigger sequence

Reviewer #4 (Remarks to the Author):

The authors have responded effectively - thoroughly and reasonably convincingly - to the many questions and comments induced by the original draft.

The new version is more clear in presenting reliable data and their interpretation. Conclusions are reasonable and data-based. Discussion is a little less tight than in original MS, undoubtedly because the many reviewers' issues that have been addressed necessarily result in a lot of new narrative.

This paper is likely to support and stimulate important research, including research aimed at major clinical phenomena, such as arterial thrombosis and platelet transfusion/cell therapy.

Response to the Reviewers, round 2

Please see below our response (in *italics*) to the reviewers' comments. The changes made to the manuscript are colored in red.

Reviewers' comments:

Reviewer #1 (Remarks to the Author):

I thank the authors for their clear responses to my questions and applaud the major changes made in the manuscript. I congratulate the authors with very interesting data. I have no additional comments.

Response 1: We appreciate this reviewer's assessment and thank him/her for spending time reviewing our work and providing comments that have helped to strengthen our manuscript.

Reviewer #2 (Remarks to the Author):

In rebuttal letter, the authors argue that “Filopodia formation in platelets and transfected CHO cells expressing GPIb-IX has been used to indicate GPIb-IX signaling for more than 15 years and reproduced in several reputable independent labs”.

However, receptor signaling should be indicated by the biochemical/biophysical changes induced by the ligand binding to the receptor, resulting in physiological/pathological relevant cellular responses. Hence to conclude that a receptor transduces signal, one should (1) show that the event is induced by a ligand binding to the receptor, (2) the event is not caused by structural changes to cellular model system (such as that caused by mutations etc), and (3) the event is relevant to certain physiological cell responses. In the previous version, the authors did not provide evidence the membrane morphology changes (filopodia) they saw were caused by the receptor function of GPIb-IX, did not exclude the possibility of structural changes to the membranes, and did not provide a physiological signaling outcome of GPIb-IX signaling, such as integrin activation. Please note that GPIb-IX is also a structural membrane protein, which is tightly associated the cytoskeletal proteins critically important for cell membrane integrity and morphology. It is more than a possibility that a mutation in the GPIb directly affects morphology of the membrane without requiring receptor (or signaling) function of GPIb-IX (mutations or deletion in GPIb-IX in Bernard-Soulier syndrome causes dramatic platelet membrane structure and morphology changes, but one cannot conclude that it is caused by receptor signaling of GPIb-IX. Therefore, the authors' argument is not convincing.

Response 2: This reviewer listed 3 criteria for receptor signaling and claimed that our filopodia data did not meet any of them. We respectfully disagree with this claim.

About his first claim that we “did not provide evidence ... (filopodia) they saw were caused by the receptor function of GPIb-IX”: we showed in Figure 4c and 4d that GPIb-IX expressed in CHO cells mediated cell attachment to the VWF surface, clearly establishing filopodia formation

as a consequence of the receptor function of GPIb-IX. Moreover, our results were consistent with earlier studies as cited in the manuscript. Additional control experiments (e.g. usage of anti-LBD antibodies that block VWF binding to GPIba) had been performed in these earlier studies to establish the filopodia assay. This reviewer quoted our response citing earlier studies, but he seemed to have ignored these studies and our consistent results. It would be unreasonable if he ignored Figure 4d and yet expected us to perform more control experiments that had already been reported in earlier studies.

About his second claim that we “did not exclude the possibility of structural changes to the membranes”: I am not sure what this reviewer meant by “structural changes to the membranes”. But pertaining to his second criterion of receptor signaling, the filopodia was observed in the wild-type platelet and in CHO cells expressing wild-type GPIb-IX. Filopodia produced in CHO cells expressing mutant GPIba Δ was of similar shape and length to those in wild-type cells (Fig. 4e).

This reviewer mentioned Bernard-Soulier syndrome (BSS), seeming to imply that we did not exclude the possibility that mutant GPIba Δ caused filopodia by causing BSS-like phenomenon. This is misleading. BSS is caused by the lack of expression of functional GPIb-IX on the platelet surface. In comparison, filopodia is caused by GPIb-IX signaling. BSS is due to a mutation in GPIb-IX that disrupts its expression or complex assembly. We have clearly excluded this possibility by reporting in the manuscript that “GPIba Δ assembly with GPIb β and GPIX and its interaction with A1 are wild type-like” and citing our earlier work (ref #14) in which the wild type-like expression level of GPIba Δ was presented. Moreover, although BSS platelets are much larger than normal ones and exhibited an altered open canalicular membrane system, this kind of morphology change is very different from the filopodia observed in our study. Morphology wise, CHO cells expressing BSS-causing GPIb-IX look essentially the same as CHO cells not expressing GPIb-IX and therefore do not bind VWF and produce filopodia. Therefore, BSS is distinct from and not related to the filopodia formation. It is very curious to me that this reviewer chose to use similar terms (“membrane morphology changes” and “membrane structure and morphology changes”) to describe two different and unrelated phenomena (filopodia and changes in BSS platelets).

About his third claim that we “did not provide a physiological signaling outcome of GPIb-IX signaling, such as integrin activation”: we do not agree with his claim. As described in earlier studies, filopodia are observed in both platelets and CHO cells expressing GPIb-IX. Moreover, in addition to the filopodia formation in IL4R-IbaTg platelets, we showed in the revised manuscript that these mutant platelets exhibited other signaling changes such as increased intracellular calcium concentration and increased P-selectin expression (Fig. 6d), both of which are well-documented physiological events. Thus, using these indicators of signaling, I believe we have provided sufficient evidence to support the idea that MSD unfolding induces GPIb-IX signaling.

This reviewer appeared to be invested with the notion that GPIb signaling is sufficient to lead to integrin activation. Contrary to what this reviewer claimed, this notion is not fully settled. For instance, many anti-LBD antibodies bind the same ligand-binding domain of GPIba as VWF does, yet antibody-induced platelet aggregation traces are clearly different from ristocetin-

induced ones. Such difference has not been fully explained. Thus, for our study integrin activation may not be an optimal indicator of GPIb-IX signaling.

With that being said, the authors in this revision have provided a new figure showing a monoclonal antibody against GPIIb/IIIa (RAM1), as previously reported, diminished filopodia in GPIb-IX-expressing cells, which is an improvement. However, GPIIb/IIIa is not the receptor subunit of GPIb-IX. A vWF binding blocking anti-GPIb antibody will be much more convincing. Also, the possibility of RAM1 affected GPIb-IX cytoskeletal association or membrane structure has not been excluded here or in the original RAM1 paper in ATVB (which was not a convincing paper by any standard). Thus the authors should use the receptor blocking anti-GPIb antibody and show that filopodia occurs only on vWF but not other matrix proteins such as fibronectin or fibrinogen.

Response 3: It has already been reported in earlier studies, and verified by us (unpublished data), that anti-LBD antibodies in the presence of EDTA blocked the attachment of platelets or CHO cells to the VWF surface. With no attached cells, it would be impossible to visualize the filopodia in cells. To demonstrate filopodia occurs only on VWF, we have showed that cells adhered to VWF only in the presence of botrocetin and when GPIb-IX is expressed (Fig. 4c,d).

Similarly, it has already been reported in earlier studies that anti-LBD antibodies in the absence of EDTA did not block attachment of cells to the VWF surface because the cells are attached to VWF through integrins. When integrins are engaged, cells spread over the VWF surface and form lamellipodia, again making it impossible to visualize the filopodia in cells. (That is why EDTA was used in our assay to block integrin signaling.) This reviewer's suggestion of using fibronectin or fibrinogen is not good, since both fibronectin and fibrinogen would induce integrin-mediated cell spreading and lamellipodia formation.

I don't understand why this reviewer thinks we need to exclude the possibility of RAM.1 affecting GPIb-IX cytoskeletal association. That RAM.1 inhibits filopodia formation in iCHO-Iba Δ /Ib β /IX cells indicates that the effect of GPIIb/IIIa mutation propagates through GPIb-IX, instead of an unrelated receptor, into the cell since RAM.1 binds to the same complex. This conclusion would not be impacted whether or not RAM.1 affects GPIb-IX cytoskeletal association.

Because the goal of this manuscript is to study the receptor signaling mechanisms and the role of MSD in transducing such a signal, it is important to exclude constitutive changes caused by mutations or antibodies that affect membrane morphology. Unfortunately, the only positive data on signaling this manuscript present are the constitutive morphology changes.

Response 4: This reviewer might have overlooked the data we added in the last revision, particularly the increased intracellular calcium and increased P-selectin expression in the IL4R-IbaTg platelets (Fig. 6d). These additional "positive data" support the conclusion of constitutive GPIb-IX signaling in these mutant platelets. Also, it should be noted that these constitutive signaling events in IL4R-IbaTg platelets are consistent with those in wild-type platelets induced by the botrocetin/shear treatment.

In previous submission, the reviewer commented that "the data suggesting GPIb-IX MSD

mutant cells and IL4R-Iba-tg platelets induce constitutive signaling independent of VWF binding can be explained by too many other possibilities.” In response, the authors state that “it is difficult to address this broad critique, but I hope the aforementioned RAM.1 inhibition of filopodia formation in iCHO-Ib α Δ /Ib β /IX cells would help to allay this reviewer's suspicions.”

The “too many possibilities” comment is based on the reasons explained above. Because the authors did not provide data supporting a receptor-based signaling, it is entirely possible that all kinds of structural changes induced by a mutation can be responsible for the morphological changes. For the same reason above, the conclusion of constitutive IL4-GPIb fusion protein signaling is not convincing.

Response 5: Thanks for this reviewer for clarifying his comment of “too many possibilities”. As responded above, we respectfully disagree with his opinion on the constitutive signaling in IL4R-IbaTg platelets.

In fact, IL4-GPIb clearly abolished GPIb-associated functions in previous studies by investigators who made the mutant mice.

Response 6: Dr. Jerry Ware, who made the IL4R-IbaTg mice, is a coauthor on this manuscript. He agrees with the conclusion of this manuscript!

Not all “GPIb-associated functions” are “abolished” in this transgenic mouse. Nor are they fully explained. For instance, although Dr. Ware noted in his earlier paper that the mice were thrombocytopenic, the underlying mechanism was not addressed. In this manuscript we showed for the first time that these transgenic platelets are cleared faster than the wild-type ones and we provided a mechanism for it.

In response 17, the authors stated “Whether the MSD is important for integrin activation is beyond the scope of this manuscript.” If the data supporting signaling is convincing, this reply may be ok, but when the morphological changes are not convincing to the reviewer, showing whether MSD is important for integrin activation become an easy way to convince the reviewer that it is indeed mediating signaling. In fact, the reviewer was suggesting how to improve the manuscript.

Response 7: Thanks for his suggestion of utilizing integrin activation as an indicator of GPIb-IX signaling in our study. As described above, we feel that integrin activation may not be an optimal indicator of GPIb-IX signaling. Instead, we have utilized other indicators such as increased intracellular calcium concentration and P-selectin expression.

Reviewer #3 (Remarks to the Author):

Overall I am satisfied with the revisions and this paper does give a new way of viewing the function of a key platelet receptor and using a diverse array of techniques links the receptor unfolding mechanisms which has ramifications for understanding the widespread phenomenon of platelet clearance and also important for developing new ways of platelet

storage.

Response 8: We appreciate this reviewer's assessment and thank him/her for spending time reviewing our work and providing comments that have helped to strengthen our manuscript.

minor correction

Page 15. Line 324 – insert “the”

These results also suggest that the extension of “the” Trigger sequence

Response 9: “the” was added as suggested. Also, we have gone through the entire manuscript and corrected similar mistakes.

Reviewer #4 (Remarks to the Author):

The authors have responded effectively – thoroughly and reasonably convincingly – to the many questions and comments induced by the original draft.

The new version is more clear in presenting reliable data and their interpretation. Conclusions are reasonable and data-based. Discussion is a little less tight than in original MS, undoubtedly because the many reviewers' issues that have been addressed necessarily result in a lot of new narrative.

This paper is likely to support and stimulate important research, including research aimed at major clinical phenomena, such as arterial thrombosis and platelet transfusion/cell therapy.

Response 10: We appreciate this reviewer's assessment and thank him/her for spending time reviewing our work and providing comments that have helped to strengthen our manuscript.

REVIEWERS' COMMENTS:

Reviewer #2 (Remarks to the Author):

The hypothesis of the manuscript is a good one. However, it is disappointing that the authors made the toned arguments without adding more convincing evidence to support their conclusions, which would have significantly improved the quality of the work. The reviewer is not against the idea that GPIb mediates signals, such as that shown in Fig 1 (although the conclusion of that figure is not new). However, the point of the manuscript is to show that MSD structure is important for transmitting VWF binding-dependent shear force to induce signals, for which evidence presented is not sufficient. The authors have made a deletion mutant of GPIba removing the sequences from the "MSD" region, and shown that the mutant-expressing cells display more filopodia-like structure in regions away from VWF contact area, as compared to wild type GPIb-expressing cells, which show similar structure in the VWF contact area (like most adherent cells). These data suggest that mutations altered GPIb in a way that causes these morphological changes. As the authors are aware of, GPIb is critically important for maintaining membrane morphology in platelets. mutations in GPIb or reduction in GPIb expression levels may have the potential to cause morphological changes (an extreme example is the BSS platelets, which has no GPIb). It is necessary to exclude the possibility that MSD deletion or GPIB extracellular domain deletion simply disturbs a normal functional state of wild type GPIb to maintain the membrane structure (more subtly than totally lacking GPIb in BSS) in order to suggest that the morphology changes caused by deletion mutations is due to the activation of GPIb signaling. In fact, if MSD is important in such "signaling", deletion of MSD should have abolished VWF-induced "filapodia"-like morphology, instead of causing constitutive filapodia-like structure in places where cells are not in contact with VWF.

Additionally, platelets lacking GPIb expression have a tendency to be "more active" than platelets expressing normal GPIb mainly due to the dramatic changes in membrane structure. Does the GPIb ko platelets show "constitutive" calcium elevation as compared to wild type.

Response to the Reviewers, round 3

Please see below our response (in *italics*) to the reviewers' comments. The changes made to the manuscript are tracked.

Reviewer #2 (Remarks to the Author):

The hypothesis of the manuscript is a good one. However, it is disappointing that the authors made the toned arguments without adding more convincing evidence to support their conclusions, which would have significantly improved the quality of the work. The reviewer is not against the idea that GPIb mediates signals, such as that shown in Fig 1 (although the conclusion of that figure is not new). However, the point of the manuscript is to show that MSD structure is important for transmitting VWF binding-dependent shear force to induce signals, for which evidence presented is not sufficient. The authors have made a deletion mutant of GPIba removing the sequences from the "MSD" region, and shown that the mutant-expressing cells display more filopodia-like structure in regions away from VWF contact area, as compared to wild type GPIb-expressing cells, which show similar structure in the VWF contact area (like most adherent cells). These data suggest that mutations altered GPIb in a way that causes these morphological changes. As the authors are aware of, GPIb is critically important for maintaining membrane morphology in platelets. mutations in GPIb or reduction in GPIb expression levels may have the potential to cause morphological changes (an extreme example is the BSS platelets, which has no GPIb). It is necessary to exclude the possibility that MSD deletion or GPIB extracellular domain deletion simply disturbs a normal functional state of wild type GPIb to maintain the membrane structure (more subtly than totally lacking GPIb in BSS) in order to suggest that the morphology changes caused by deletion mutations is due to the activation of GPIb signaling. In fact, if MSD is important in such "signaling", deletion of MSD should have abolished VWF-induced "filapodia"-like morphology, instead of causing constitutive filapodia-like structure in places where cells are not in contact with VWF.

Response 1: The purpose of Figure 1 is not to reinvent the wheel, but to demonstrate that our experimental system (i.e. cone-and-plate viscometer coupled with flow cytometry) could produce results consistent with previous studies. Thus, this experimental system could be utilized to demonstrate something entirely new (i.e. ligand/shear-induced unfolding of MSD on the platelet surface). At the same time, Figure 1 serves to remind the readers of the requirement of shear for GPIb-IX signaling, which may have been neglected in some prior studies.

This reviewer appears to misunderstand the conclusion of our study. In the GPIba Δ mutant, only a portion of MSD, not the entire domain, is deleted. On page 10 of our revised manuscript we had devoted an entire paragraph to describe how we interpreted the results of the GPIba Δ mutant – the purpose of MSD unfolding is to expose the Trigger sequence therein, which presumably will be in contact with the nearby GPIb β and GPIX subunits. The Trigger sequence is present and exposed in both GPIba Δ and IL4R-Iba constructs; we believe that's why both constructs exhibit constitutive GPIb-IX signaling. Indeed, deletion of the Trigger sequence in GPIba Δ abolished the filopodia formation in CHO-Iba Δ /Ib β /IX cells (our unpublished results).

The Trigger sequence is clearly labeled in Figures 4 and 6, included in the abstract, described extensively in the manuscript, and mentioned numerous times in my earlier response to comments of the other reviewers. However, to make it even clearer, we have revised a sentence in the first paragraph of Discussion. It now reads: “Consequently, the Trigger sequence becomes extended and presumably exposed to nearby GPIIb β and GPIX extracellular domains, ...”

Additionally, platelets lacking GPIIb expression have a tendency to be "more active" than platelets expressing normal GPIIb mainly due to the dramatic changes in membrane structure. Does the GPIIb ko platelets show "constitutive" calcium elevation as compared to wild type.

Response 2: I don't know the answer to this reviewer's question; our study utilized neither GPIIb $\alpha^{-/-}$ nor GPIIb $\beta^{-/-}$ mice. I am not sure whether the dramatic changes in the membrane structure of platelets lacking GPIIb (i.e. Bernard-Soulier platelets) are directly related to what we are studying in this manuscript, considering the pleiotropic nature of the Bernard-Soulier platelets. Also, I am not aware of any publications showing that Bernard-Soulier platelets are “more active” than the wild-type, and in similar manners as those induced by the botrocetin/VWF/shear treatment described here.